# Two NLR immune receptors acquired high-affinity binding to a fungal effector through convergent evolution of their integrated domain

Aleksandra Białas[1], Thorsten Langner[1], Adeline Harant[1], Mauricio P Contreras[1], Clare EM Stevenson[2], David M Lawson[2], Jan Sklenar[1], Ronny Kellner[1], Matthew J Moscou[1], Ryohei Terauchi[3,4], Mark J Banfield[2], Sophien Kamoun[1]*

[1]The Sainsbury Laboratory, University of East Anglia, Norwich Research Park, Norwich, United Kingdom; [2]Department of Biological Chemistry, John Innes Centre, Norwich Research Park, Norwich, United Kingdom; [3]Division of Genomics and Breeding, Iwate Biotechnology Research Centre, Iwate, Japan; [4]Laboratory of Crop Evolution, Graduate School of Agriculture, Kyoto University, Kyoto, Japan

**Abstract** A subset of plant NLR immune receptors carry unconventional integrated domains in addition to their canonical domain architecture. One example is rice Pik-1 that comprises an integrated heavy metal-associated (HMA) domain. Here, we reconstructed the evolutionary history of Pik-1 and its NLR partner, Pik-2, and tested hypotheses about adaptive evolution of the HMA domain. Phylogenetic analyses revealed that the HMA domain integrated into Pik-1 before Oryzinae speciation over 15 million years ago and has been under diversifying selection. Ancestral sequence reconstruction coupled with functional studies showed that two Pik-1 allelic variants independently evolved from a weakly binding ancestral state to high-affinity binding of the blast fungus effector AVR-PikD. We conclude that for most of its evolutionary history the Pik-1 HMA domain did not sense AVR-PikD, and that different Pik-1 receptors have recently evolved through distinct biochemical paths to produce similar phenotypic outcomes. These findings highlight the dynamic nature of the evolutionary mechanisms underpinning NLR adaptation to plant pathogens.

*For correspondence:
sophien.kamoun@tsl.ac.uk

## Introduction

*N*ucleotide-binding domain *l*eucine-rich *r*epeat–containing (NLR) proteins constitute an ancient class of intracellular immune receptors that confer innate immunity in plants and animals (*Dodds and Rathjen, 2010*; *Jones et al., 2016*). In plants, NLRs function by sensing pathogen-derived virulence molecules, known as effectors, and subsequently activating an immune response (*Jacob et al., 2013*; *Kourelis and van der Hoorn, 2018*). The majority of functionally validated NLRs in plants display broadly conserved domain architectures, typically consisting of the NB-ARC (*n*ucleotide-*b*inding *a*daptor shared by APAF-1, certain *R* gene products and *CED-4*) domain, the LRR (*leucin-rich repeat*) region, and either a TIR (*Toll/interleukin 1 receptor*), CC (*coiled-coil*), or CC_R (RPW8-type CC) domain at the N-terminus (*Kourelis and Kamoun, 2020*; *Shao et al., 2016*). However, coevolution with pathogen effectors has led to a remarkable diversification of NLR repertoires, which form one of the most diverse protein families in plants (*Lee and Chae, 2020*; *Prigozhin and Krasileva, 2020*). An emerging paradigm in plant immunity is that some NLRs acquired novel recognition specificities through fusions of noncanonical integrated domains (IDs) that mediate perception of effectors (*Cesari et al., 2014a*; *Wu et al., 2015*). Although NLR-IDs have been described across various plant families (*Gao et al., 2018*; *Kroj et al., 2016*; *Sarris et al., 2016*; *Van de Weyer et al., 2019*), little is

known about their emergence and subsequent evolution. In addition, our knowledge about how NLRs adapt to rapidly evolving pathogen effectors remains sparse.

Given that many IDs exhibit homology to molecules required for immune responses, they are generally thought to have derived from effector operative targets, which then act as baits for effector recognition within NLRs (*Cesari et al., 2014a*; *Wu et al., 2015*). IDs can perceive effectors by direct binding, by serving as substrates for their enzymatic activities, or by detecting effector-induced perturbations (*Bao et al., 2017*; *Cesari et al., 2014a*; *Fujisaki et al., 2017*; *Heidrich et al., 2013*; *Sarris et al., 2016*; *Wu et al., 2015*). The RGA5 (also known as Pia-2) and Pik-1 receptors are well-characterised examples of NLR-IDs. RGA5 and Pik-1 detect three unrelated effectors from the rice blast fungus, *Magnaporthe oryzae*, AVR-Pia/AVR1-CO39 and AVR-Pik, respectively, via their integrated heavy metal-associated (HMA) domains (*De la Concepcion et al., 2018*; *Guo et al., 2018*). HMAs are commonly found in a family of *H*MA *p*lant *p*roteins (HPPs) or *H*MA *i*soprenylated *p*lant *p*roteins (HIPPs) known to contribute to abiotic and biotic stress responses (*de Abreu-Neto et al., 2013*; *Fukuoka et al., 2009*; *Li et al., 2020a*; *Radakovic et al., 2018*; *Zschiesche et al., 2015*). Recently, the AVR-Pik effectors have been shown to bind and stabilise rice HMA proteins to co-opt their function in immunity (*Maidment et al., 2020*; *Oikawa et al., 2020*), providing direct evidence that integrated HMAs indeed mimic host targets of effectors.

NLR-triggered immunity is usually accompanied by the *h*ypersensitive *r*esponse (HR), a type of localised cell death associated with disease resistance. Notably, several NLR-IDs appear to have lost the ability to autonomously trigger a defence response (*Cesari et al., 2014b*; *Zdrzałek et al., 2020*). As a consequence, they often function in pairs, where the NLR-ID serves as a sensor for pathogen effectors and its partner acts as a helper that mediates activation of an immune response (*Adachi et al., 2019*; *Bonardi et al., 2011*; *Feehan et al., 2020*). There are now many examples of such NLR pairs, including RRS1/RPS4 from *Arabidopsis thaliana* (*Saucet et al., 2015*) as well as Pik-1/Pik-2 (*Ashikawa et al., 2008*), Pii-2/Pii-1 (*Fujisaki et al., 2017*), and RGA5/RGA4 (the *Pia* locus) (*Cesari et al., 2014b*; *Okuyama et al., 2011*) from rice. Many NLR pairs are encoded by two adjacent genes in a head-to-head orientation (*Bailey et al., 2018*; *Van de Weyer et al., 2019*). This genetic linkage likely provides an evolutionary advantage by facilitating co-segregation, coevolution, and transcriptional coregulation of functionally linked genes (*Baggs et al., 2017*; *Griebel et al., 2014*). Genetic linkage may also reduce the genetic load caused by autoimmunity (*Wu et al., 2018*), which is a common phenomenon observed across NLRs (*Alcázar et al., 2009*; *Bomblies et al., 2007*; *Chae et al., 2016*; *Deng et al., 2019*; *Yamamoto et al., 2010*).

Rice Pik-1 and Pik-2 proteins form a CC-type NLR pair. Two *Pik* haplotypes, N- and K-type, are present in the genetic pool of wild and cultivated rice (*Zhai et al., 2011*). While the function of the N-type haplotypes remains obscure, K-type Pik NLRs confer resistance to the rice blast fungus. In the K-type pair, Pik-1 acts as a sensor that binds the AVR-Pik effector via the Pik-1-integrated HMA domain, whereas Pik-2 is required for activation of immune response upon effector recognition (*Maqbool et al., 2015*; *Zdrzałek et al., 2020*). This NLR pair was initially cloned from Tsuyuake rice (*Ashikawa et al., 2008*) and has since been shown to occur in allelic variants, which include Pikp, Pikm, Piks, Pikh, and Pik* (*Costanzo and Jia, 2010*; *Jia et al., 2009*; *Wang et al., 2009*; *Yuan et al., 2011*; *Zhai et al., 2011*). Remarkably, the integrated HMA domain is the most sequence-diverse region among Pik-1 variants, consistent with the view that the receptor is under selection imposed by AVR-Pik (*Białas et al., 2018*; *Costanzo and Jia, 2010*; *De la Concepcion et al., 2021*; *Zhai et al., 2014*). Conversely, AVR-Pik alleles carry only five amino acid replacements, all of which map to regions located at the HMA-binding interface, indicating the adaptive nature of those polymorphisms (*Longya et al., 2019*). While the most ancient of the AVR-Pik allelic variants, AVR-PikD, is recognised by a wide range of Pik-1 proteins, the most recent variants, AVR-PikC and AVR-PikF, evade recognition by all known Pik-1 variants (*Kanzaki et al., 2012*; *Longya et al., 2019*). These recognition specificities are thought to reflect the ongoing arms race between rice and the rice blast fungus (*Białas et al., 2018*; *Kanzaki et al., 2012*; *Li et al., 2019*) and have been linked to the effector–HMA binding affinity (*De la Concepcion et al., 2021*; *De la Concepcion et al., 2018*; *Maqbool et al., 2015*). Despite the wealth of knowledge about mechanisms governing effector recognition by the Pik-1-integrated HMA domain, we know little about its evolutionary history.

Evolutionary molecular biology can inform mechanistic understanding of protein function. After decades of parallel research, molecular evolution and mechanistic research are starting to be used in conjunction to unravel the molecular basis of protein function within an evolutionary framework

(*Delaux et al., 2019*). One approach to investigate the biochemical drivers of adaptation is to reconstruct the evolutionary trajectories of proteins of interest (*Dean and Thornton, 2007*; *Harms and Thornton, 2013*; *Thornton, 2004*). Using phylogenetic techniques and algorithms for ancestral *sequence reconstruction* (ASR), it is now possible to statistically infer ancestral sequences, which can then be synthesised, expressed, and experimentally studied in the context of modern sequences (*Ashkenazy et al., 2012*; *Cohen and Pupko, 2011*; *Pupko et al., 2000*). In the field of plant–microbe interactions, experimental analyses of resurrected ancestral effector sequences have helped unravel biochemical bases of effector specialisation and adaptive evolution following a host jump (*Dong et al., 2014*; *Tanaka et al., 2019*; *Zess et al., 2019*). To date, ancestral reconstruction has not been used to study the evolution of NLR immune receptors.

Despite remarkable advances in the field of NLR biology, there is still a significant gap in our understanding of how these receptors have adapted to fast-evolving pathogens. In this work, we used the rice Pik-1/Pik-2 system, coupled with ASR, to test hypotheses about adaptive evolution of NLRs and their IDs and to bridge the gap between mechanistic and evolutionary research. We leveraged the rich genetic diversity of the *Pik* genes in grasses and discovered that they likely derived from a single ancestral gene pair that emerged before the radiation of the major grass lineages. In addition, we show that the HMA integration predates speciation of Oryzinae dated at ~15 million years ago (MYA) (*Jacquemin et al., 2011*; *Stein et al., 2018*). Functional characterisation of a resurrected ancestral HMA (ancHMA), dating back to early *Oryza* evolution, revealed that different allelic variants of Pik-1, Pikp-1 and Pikm-1, convergently evolved from the weakly binding ancestral state towards high-affinity binding and recognition of the AVR-PikD effector through different biochemical paths. We conclude that for most of its evolutionary history Pik-HMA did not sense AVR-PikD and that recognition of this effector is a recent adaptation. This work provides new insights into our understanding of the dynamic nature of NLR adaptive evolution.

## Results

### Pik orthologues are widely present across distantly related grass species

To determine the diversity of the *Pik-1* and *Pik-2* genes across the Poaceae family (grasses), we performed a phylogenetic analysis of the entire repertoire of CC-NLRs from representative grass species. We used NLR-Parser (*Steuernagel et al., 2015*) to identify NLR sequences from publicly available protein databases of eight species (*Supplementary file 1A*). Following rigorous filtering steps (described in Materials and methods), we compiled a list of 3062 putative CC-NLRs (*Supplementary file 2*), amended with known and experimentally validated NLR-type proteins from grasses (*Supplementary file 1B*). Next, we constructed a *maximum likelihood* (ML) phylogenetic tree based on protein sequences of the NB-ARC domain of recovered CC-NLRs and discovered that the Pik-1 and Pik-2 sequences fell into two phylogenetically unrelated, but well-supported, clades (*Figure 1—figure supplement 1A*). Among Pik-1- and Pik-2-related sequences, we detected representatives from different, often distantly related, grass species, including members of the Pooideae and Panicoideae subfamilies. To determine the topologies within these clades, we performed additional phylogenetic analyses using codon-based sequence alignments of Pik-1 and Pik-2 clade members. Both Pik-1 and Pik-2 phylogenetic trees, calculated using the ML method, revealed the relationships within the two clades (*Figure 1—figure supplement 1B*). We propose that the identified clades consist of Pik-1 and Pik-2 orthologues from a diversity of grass species.

We noted that Pik-2 from *Oryza brachyantha* was N-terminally truncated as a result of a 46 bp deletion within its 5′-region (*Figure 1—figure supplement 2A*). To determine whether the *O. brachyantha* population carries a full-length *Pik-2* gene, we genotyped 16 additional *O. brachyantha* accessions (*Figure 1—figure supplement 2B*). We successfully amplified and sequenced six full-length *ObPik-2* genes, none of which carried the deletion present in the reference genome. We further amplified full-length *ObPik-1* genes from the selected accessions (*Supplementary file 1C*), confirming that both full-length *Pik-2* as well as *Pik-1* are present in this species.

Following these results, we expanded the search of Pik orthologues to 10 additional species, focusing on members of the Oryzoideae subfamily (*Supplementary file 1D*). Using recurrent BLASTN searches combined with manual gene annotation and phylogenetic analyses, we identified

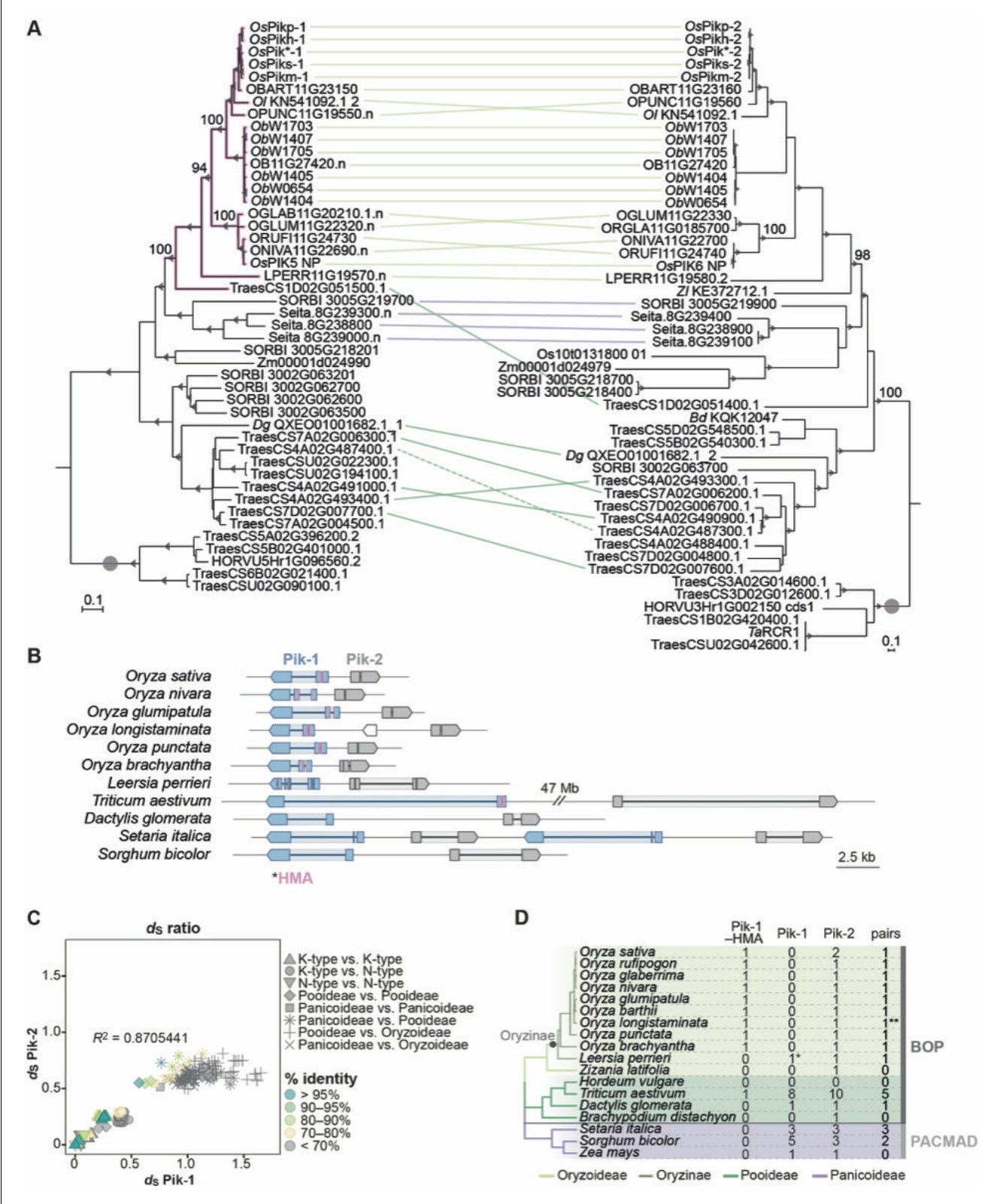

**Figure 1.** The *Pik-1*/*Pik-2* orthologues are distributed across diverse species of grasses. (**A**) The *m*aximum likelihood (ML) phylogenetic trees of Pik-1 (left) and Pik-2 (right) orthologues. The trees were calculated from 927- and 1239-nucleotide-long codon-based alignments of the NB-ARC domain, respectively, using RAxML v8.2.11 (*Stamatakis, 2014*), 1000 bootstrap method (*Felsenstein, 1985*), and GTRGAMMA substitution model (*Tavaré, 1986*). Best ML trees were manually rooted using the selected clades (marked with grey circles) as outgroups. The bootstrap values above

*Figure 1 continued on next page*

*Figure 1 continued*

70% are indicated with grey triangles at the base of respective clades; the support values for the relevant nodes are depicted with numbers. The scale bars indicate the evolutionary distance based on nucleotide substitution rate. The Pik-1 integration clade is shown in pink. Genetically linked genes are linked with lines, with colours indicating plant subfamily: Oryzoideae (purple), Pooideae (dark green), or Panicoideae (light green); the continuous lines represent linkage in a head-to-head orientation, the dashed line indicates linkage in a tail-to-tail orientation. The interactive trees are publicly available at: https://itol.embl.de/tree/14915519290329341598279392 and https://itol.embl.de/tree/14915519290161451596745134. (**B**) Schematic illustration of the *Pik* locus in selected species. The schematic gene models of *Pik-1* (blue) and *Pik-2* (grey) are shown. The integrated heavy metal-associated (HMA) domain is marked with pink. The coordinates of the regions presented in this figure are summarised in *Supplementary file 1E*. (**C**) Comparisons of pairwise $d_S$ rates calculated for the Pik-1 and Pik-2 receptors. The rates were calculated using *Yang and Nielsen, 2000* based on 972- and 1269-nucleotide-long codon-based alignments of the NB-ARC domains of Pik-1 and Pik-2, respectively; only positions that showed over 70% coverage across the alignment were used for the analysis. The comparisons were categorised to reflect species divergence (shapes) and colour-coded to illustrate percentage identity of $d_S$ values (% identity). The coefficient of determination ($R^2$) was calculated for each dataset using R v3.6.3 package. (**D**) Summary of identified Pik-1 and Pik-2 homologues in plant species included in this study. The phylogenetic tree was generated using TimeTree tool (*Kumar et al., 2017*). The number of pairs correspond to the number of *Pik-1/Pik-2* genes in a head-to-head orientation separated by intergenic region of various length. **The species harbours a truncated gene between *Pik-1* and *Pik-2*; *the species has likely lost the HMA domain; Pik-1–HMA: Pik-1 with the HMA domain; Pik-1: Pik-1 without the HMA integration; BOP: *Bambusoideae, Oryzoideae, Pooideae*; PACMAD: *Panicoideae, Arundinoideae, Chloridoideae, Micrairoideae, Aristidoideae, Danthonioideae*.

The online version of this article includes the following source data and figure supplement(s) for figure 1:

**Source data 1.** Selection test for Pik-1 vs. Pik-2 orthologues.
**Figure supplement 1.** Pik-1 and Pik-2 orthologues fall into two well-supported clades.
**Figure supplement 2.** Genotyping of *Oryza brachyantha* accession.
**Figure supplement 3.** Pik-1 and Pik-2 orthologues from *Oryza* spp. fall into K- and N-type clades.
**Figure supplement 4.** Schematic representation of selected *Pik* clusters in wheat (*T. aestivum*), sorghum (*S. bicolor*), and foxtail millet (*S. italica*).
**Figure supplement 5.** Random pairwise comparisons of $d_S$ rates calculated for the Pik-1 and Pik-2 receptors.
**Figure supplement 6.** Genetically linked Pik-1 and Pik-2 have similar molecular age.

additional Pik-related NLRs resulting in 41 and 44 Pik-1 and Pik-2 sequences, respectively (*Figure 1A*). Altogether, the additional Pik orthologues gave us a broad view of their occurrence in monocots. The majority of species within the Oryzinae subtribe contain single copies of *Pik-1* and *Pik-2* per accession, whereas members of the Pooideae and Panicoideae subfamilies frequently encode multiple Pik-1 or Pik-2 paralogues, with wheat carrying as many as 9 and 10 *Pik-1* and *Pik-2* genes, respectively. It is possible that ancestral *Pik-1* and *Pik-2* experienced a duplication before the radiation of Pooideae and Panicoideae followed by different patterns of gene loss/retention among grass species; however, a better-resolved phylogeny is needed to test this possibility. In addition, Pik-1 and Pik-2 from the *Oryza* genus formed two subclades, corresponding to the two haplotypes previously identified at the *Pik* locus, N-type and K-type (*Figure 1—figure supplement 3*; *Zhai et al., 2011*). We conclude that the N- and K-type *Pik* genes have been maintained through speciation and coexist as haplotypes in different *Oryza* species. Altogether, we discovered that Pik-1 and Pik-2 orthologues are present across a wide range of grasses, including members of the Oryzoideae, Pooideae, and Panicoideae subfamilies.

## Genetic linkage of the *Pik* gene pair predates the split of major grass lineages

In rice, the *Pikp-1* and *Pikp-2* genes are located in a head-to-head orientation at a single locus of chromosome 11, and their coding sequences are separated by an ~2.5-kb-long region (*Ashikawa et al., 2008*; *Yuan et al., 2011*). To determine whether this genetic linkage is conserved in grasses, we examined the genetic loci of retrieved *Pik-1* and *Pik-2* genes. A total of 14 out of 15 species in which both genes are present carry at least one *Pik* pair with adjacent *Pik-1* and *Pik-2* genes in a head-to-head orientation. Although the length of the genes and their intergenic regions vary between species (from ~2 kb in *O. nivara* to ~48 Mb in wheat), they exhibit largely conserved gene models. Most of the *Pik-2* orthologues feature one intron in their *nucleotide-binding domain* (NBD) region (*Ashikawa et al., 2008*) while the *Pik-1* genes typically carry one or, for the genes featuring the HMA domain, two introns (*Figure 1B*; *Supplementary file 1F*). In addition, in species that carry multiple copies of *Pik-1* or *Pik-2*, the copies are typically located in close proximity or, as in wheat, in large NLR-rich gene clusters (*Figure 1—figure supplement 4*; *Supplementary file 1F*).

Given that genomic rearrangements have been reported at the *Pik* locus (*Mizuno et al., 2020*; *Stein et al., 2018*), we could not exclude the possibility that genetic linkage of the *Pik-1/Pik-2* pair emerged more than once and is a remnant of rearrangement events. We reasoned that if the gene pair have remained genetically linked over a long evolutionary period, then they should have the same molecular age. To gain insights into the evolutionary dynamics between genetically linked Pik-1 and Pik-2 receptors, we compared their rates of synonymous substitutions ($d_S$). For this analysis, we selected representative Pik-1 and Pik-2 NLRs that are genetically linked in a head-to-head orientation from 13 species; *Lp*Pik (*Leersia perrieri*) orthologues were excluded from the analysis because their unusual gene models interfered with sequence alignments (*Figure 1B*). Next, we assessed $d_S$ within the coding sequences of the NB-ARC domain between pairwise genes using the *Yang and Nielsen, 2000* method. The rates were calculated separately for Pik-1 and Pik-2 and cross-referenced such that the pairwise values for Pik-1 were compared to the respective values for cognate Pik-2 (*Figure 1—source data 1*). The comparisons revealed strong positive correlation of $d_S$ rates ($R^2$ = 0.87, p-value=0.95) between genetically linked *Pik* genes (*Figure 1C*). This was significantly higher than observed by chance, as calculated from random Pik-1–Pik-2 cross-referencing (*Figure 1— figure supplement 5*). We conclude that the Pik-1/Pik-2 pair probably became genetically linked long before the emergence of the Oryzinae clade and prior to the split of the major grass lineages— the BOP (for *Bambusoideae, Oryzoideae, Pooideae*) and PACMAD (for *Panicoideae, Arundinoideae, Chloridoideae, Micrairoideae, Aristidoideae, Danthonioideae*) clades—which dates back to 100–50 MYA (*Hodkinson, 2018*).

## The HMA integration of Pik-1 predates the emergence of Oryzinae

To better understand the evolutionary history of Pik-1 domain architecture, we looked for signatures of HMA integration among the collection of 41 Pik-1 orthologues identified. Remarkably, the presence of an HMA domain varied among *Pik-1* genes. HMA-containing Pik-1 clustered into a single well-supported clade (herein called the Pik-1 integration clade) (*Figure 1A*). All members of the Pik-1 integration clade carry the HMA domain in the same position, between the CC and NB-ARC domains of Pik-1, and feature an intron within the HMA (*Figure 1B*). This implies that these HMA domains are likely derived from a single integration event.

Using this information, we generated a sequence alignment of selected Pik-1 orthologues to define the position of the HMA integration (*Figure 2—figure supplement 1*). We focused on comparisons of representative members of the Pik-1 integration clade and their closest relatives from *Setaria italica* and *Sorghum bicolor*. This revealed that the integration site most likely falls between the KLL and KTV residues (corresponding to residues 161–163 and 284–286 of Pikp-1); however, the exact boundaries of the integration might be slightly different, given the relatively high sequence divergence around this site among the more distantly related orthologues. We further noted that the integration site encompasses a wider region than that of functionally characterised HMA domains (*De la Concepcion et al., 2021*; *De la Concepcion et al., 2018*), with around 20 additional amino acids (23 and 21 in Pikp-1) on each side of the annotated HMA domain.

Next, we estimated when Pik-1 acquired the HMA from the phylogeny of the plant species with Pik-1 orthologues (*Figure 1D*). We found that all *Oryza* Pik-1 orthologues carry the HMA domain, which indicates that the integration predates speciation of this genus. Although we failed to detect a full-length HMA integration in *L. perrieri*, *Lp*Pik-1 carries ~15 amino acids characteristic of the HMA integration site (*Figure 2—figure supplement 1*), indicating that the fusion probably occurred before the speciation of Oryzinae, dated at ~15 MYA (*Jacquemin et al., 2011*), and was subsequently lost in *L. perrieri*. By contrast, the vast majority of examined Pik-1 from the Pooideae and Panicoideae subfamilies lack the HMA domain. The only integration in these taxonomic groups was detected in one of the nine Pik-1 paralogues of wheat included in the analysis. This observation may indicate that the Pik-1–HMA fusion may have emerged prior to radiation of the BOP clade, 100–50 MYA (*Hodkinson, 2018*). However, it is also possible that the integration occurred much later and that the newly emerged Pik-1–HMA gene transferred to wheat through introgression from rice progenitors. In summary, we can confidently conclude that the HMA integration of Pik-1 predates the emergence of the Oryzinae.

## The integrated HMA domain carries signatures of positive selection

In rice, the Pik-1-integrated HMA domain exhibits higher levels of polymorphisms compared with canonical domains of Pik-1 and Pik-2 (*Costanzo and Jia, 2010*; *Kanzaki et al., 2012*). To characterise the pressures underlying HMA diversification, we examined molecular signatures of selection within the Pik-1 integration clade. Wheat Pik-1–HMA was excluded from the analysis due to its high sequence divergence relative to *Oryza* orthologues, which precluded generating reliable sequence alignments. For the same reason, the remaining sequences were assigned into K- and N-type sequences based on phylogenetic relationship and analysed separately. To test for signatures of selection, we calculated rates of synonymous ($d_S$) and nonsynonymous ($d_N$) substitutions across the coding sequences of the HMA domain. We discovered that $d_N$ was greater than $d_S$ in 96 out of 115 pairwise sequence comparisons (86/105 for K- and 10/10 for N-type HMAs; $\omega = d_N/d_S$ ranging 0–2.45 for K-type and 1.13–3.50 for N-type) (*Figure 2A–C*), providing evidence that positive selection has acted on the integrated HMA domain. By contrast, only 9 out of 115 pairs of the NB-ARC domain sequences of the same set of genes displayed $d_N$ greater than $d_S$ (*Figure 2B–D*); however, all of these showed $d_S = 0$ and were therefore inconclusive in calculating $\omega$ ($d_N/d_S$) ratios. A comparison of the $d_N$ and $d_S$ rates between the HMA and NB-ARC domains further highlighted the elevated rates of nonsynonymous substitutions within the integrated HMA domain relative to NB-ARC (*Figure 2—figure supplement 2*). Overall, these results demonstrate that the integrated HMA domain exhibits marked signatures of positive selection in contrast to the Pik-1 NB-ARC domain.

Positive selection typically acts only on particular amino acids within a protein. Therefore, we aimed to detect sites within the integrated HMA domain that experienced positive selection using the ML method (*Yang et al., 2000*). To capture additional Pik-1-integrated HMAs, we first genotyped further wild rice species for the presence of the integration. We detected the HMA integration in 21 accessions from 13 species (*Supplementary file 1H*); 10 of those showed sufficient coverage across the entire functional region of the HMA and were used for further analysis (*Figure 2—figure supplement 1*, *Figure 2—figure supplement 3A*). We excluded the N-type HMA domains from the dataset owing to their small sample size (n = 5), which would prevent meaningful data interpretation. To detect patterns of selection within the K-type integrated HMA, we applied three pairs of ML models of codon substitution: M3/M0, M2/M1, and M8/M7 (*Yang et al., 2000*). As indicated by the *l*ikelihood *r*atio *t*ests (LRTs) and posterior probabilities, ~26% of the HMA amino acid sites likely experienced positive selection (*Figure 2—figure supplement 3B, C*; *Supplementary file 3*). As a control, we performed the same tests on the NB-ARC domain of the K-type Pik-1 sequences. Although the discrete M3 model inferred that a subset of NB-ARC amino acids might be under diversifying selection (*Figure 2—figure supplement 4*), other tests failed to detect patterns of positive selection. Based on these results, we conclude that the HMA domain exhibits strong signatures of positive selection compared with the NB-ARC domain.

## Ancestral sequence reconstruction of the Pikp-1-integrated HMA domain

To understand the evolutionary trajectory of the Pik-1-integrated HMA domain, we used representative phylogenetic trees of the K-type HMA domains to reconstruct ancHMA sequences dating to the early stages of *Oryza* genus speciation. As an outgroup, we selected HMA sequences of the integrated HMA progenitors, HPPs and HIPPs (*de Abreu-Neto et al., 2013*; *Oikawa et al., 2020*), hereafter called non-integrated HMAs, from *O. sativa* and *O. brachyantha*. To perform the reconstruction, we first tested different phylogenetic methods and focused on nodes that are well-supported in both the neighbour joining (NJ) and ML phylogenies generated from a codon-based alignment (*Figure 3—figure supplement 1*). Next, we performed the ancestral sequence prediction based on protein sequence alignment, using FastML software (*Ashkenazy et al., 2012*), which has been previously shown to infer ancestral sequences with high accuracy (*Randall et al., 2016*). Multiple reconstructions yielded multiple plausible ancHMA variants (*Figure 3—figure supplement 2*; *Supplementary file 4*). To reduce the possibility of incorrect prediction, we selected six representative well-supported sequences for further studies.

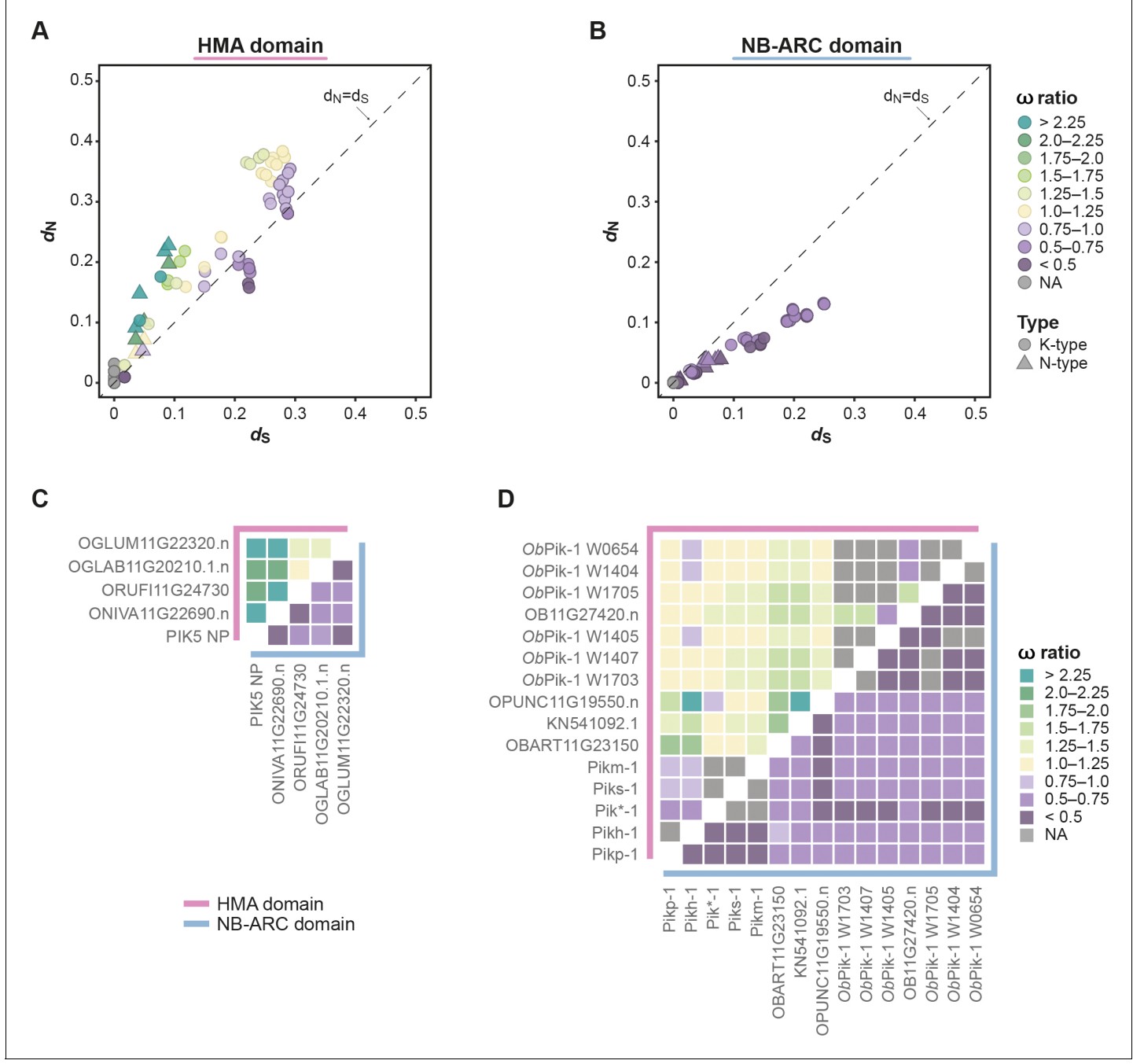

**Figure 2.** The integrated heavy metal-associated (HMA) domain exhibits elevated rates of ω ($d_N/d_S$) compared with the NB-ARC domain of Pik-1. (**A, B**) Pairwise comparison of nucleotide substitution rates within the Pik-1 integration clade for the (**A**) HMA and (**B**) NB-ARC domains, calculated using *Yang and Nielsen, 2000*. The diagonal line (dashed) indicates $d_N = d_S$. The points are colour-coded to indicate ω ratio; NA: the ratio was not calculated because $d_S = 0$. The pairwise comparisons were separately performed for the K-type (circles) and N-type (triangles) Pik-1 sequences. (**C, D**) To highlight the differences between the ω rates for the HMA (pink line) and NB-ARC (blue line) domains, the rates were plotted as heatmaps corresponding to the (**C**) N- and (**D**) K-type Pik-1 sequences.

The online version of this article includes the following source data and figure supplement(s) for figure 2:

**Source data 1.** Selection test for Pik-1-HMA vs. NB-ARC.

**Figure supplement 1.** Multiple sequence alignment illustrating the conservation around the HMA integration site.

**Figure supplement 2.** The integrated heavy metal-associated (HMA) domain displays elevated rates of $d_N$ compared with the NB-ARC domain of Pik-1.

**Figure supplement 3.** Residues within the integrated heavy metal-associated (HMA) domain are likely to have experienced positive selection.

**Figure supplement 4.** Selection test at the amino acid sites within the NB-ARC domain of the K-type *Pik-1* genes.

## Reconstructed ancHMAs exhibit weaker association with AVR-PikD compared to modern Pikp-HMA

As high-affinity binding to the effector is required for the Pik-mediated immune response (*De la Concepcion et al., 2021*; *De la Concepcion et al., 2019*; *De la Concepcion et al., 2018*; *Maqbool et al., 2015*), we hypothesised that the HMA domain of Pikp-1 (Pikp-HMA) evolved towards high-affinity binding to AVR-PikD—the most ancient of the AVR-Pik effector alleles (*Bentham et al., 2021*; *Kanzaki et al., 2012*). To test this hypothesis, we resurrected the six ancHMA variants determined above by synthesising their predicted sequences and incorporating them into the Pikp-1 receptor, generating Pikp-1:I-N2, Pikp-1:I-N6, Pikp-1:II-N11, Pikp-1:II-N12, Pikp-1:III-N11, and Pikp-1:III-N12 fusions (*Figure 3A*). We then tested their association with AVR-PikD in planta in co-immunoprecipitation (co-IP) experiments. The western blot analysis revealed that the ancHMA variants exhibited a range of association strengths with AVR-PikD (*Figure 3B*; *Figure 3—figure supplement 3*). In every case, the association with ancHMA proteins was weaker than with the present-day Pikp-HMA, indicating that binding strength has likely changed over the course of the Pikp-HMA evolutionary history. For further studies, we selected the I-N2 ancHMA variant—the last common ancestor of Pik*-1, Pikp-1, Pikh-1, Piks-1, and Pikm-1—that was predicted with high

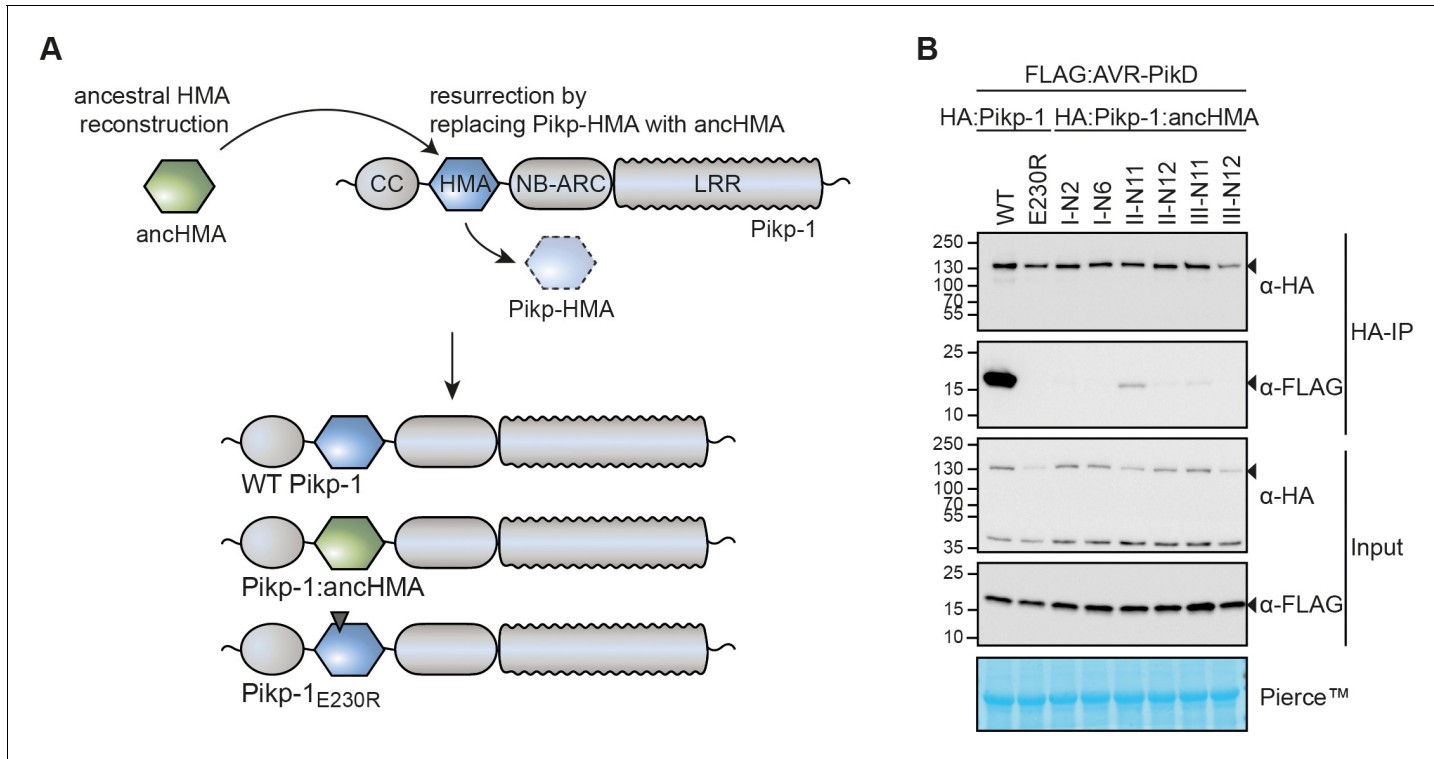

**Figure 3.** The integrated heavy metal-associated (HMA) domain of Pikp-1 exhibits stronger association with the AVR-PikD effector than its predicted ancestral state. (**A**) Overview of the strategy for resurrection of the ancestral HMA (ancHMA) domain. Following ancestral sequence reconstruction, the gene sequences were synthesised and incorporated into Pikp-1 by replacing the present-day Pikp-HMA domain (blue) with the ancHMA equivalent (green). (**B**) Co-immunoprecipitation experiment between AVR-PikD (N-terminally tagged with FLAG) and Pikp-1 (N-terminally tagged with HA) carrying ancestral sequences of the HMA. Wild-type (WT) HA:Pikp-1 and HA:Pikp-1_{E230R} were used as a positive and negative control, respectively. Immunoprecipitates (HA-IP) obtained with anti-HA probe and total protein extracts (Input) were immunoblotted with appropriate antisera (listed on the right). Rubisco loading control was performed using Pierce staining solution. Arrowheads indicate expected band sizes. Results from three independent replicates of this experiment are shown in *Figure 3—figure supplement 3*.

The online version of this article includes the following figure supplement(s) for figure 3:

**Figure supplement 1.** Phylogenetic analyses of the heavy metal-associated (HMA) domain of K-type Pik-1 NLRs.

**Figure supplement 2.** Ancestral sequence reconstruction yielded multiple plausible ancestral HMA (ancHMA) sequences.

**Figure supplement 3.** Replicates of the co-immunoprecipitation (co-IP) experiment between AVR-PikD and the reconstructed ancestral HMA (ancHMA) sequences.

confidence in probability-based method (*Supplementary file 4*) and manual accuracy assessment. The variant is called ancHMA hereafter.

## The IAQVV/LVKIE region of the Pikp-HMA domain determines high-affinity AVR-PikD binding

Next, we aimed to investigate which of the structural regions in the HMA encompass adaptive mutations towards AVR-PikD binding. By combining sequence and structural information available for Pikp-HMA (*De la Concepcion et al., 2018*; *Maqbool et al., 2015*), we identified four polymorphic regions between the ancestral and modern Pikp-HMA (*Figure 4A, B*). We sequentially replaced each of these regions in Pikp-1:ancHMA with the corresponding region from Pikp-HMA. Altogether, we obtained a suite of four chimeric HMAs—ancHMA_{AMEGNND}, ancHMA_{LVKIE}, ancHMA_{LY}, ancHMA_{PI}—and assayed these for gain-of-binding to AVR-PikD in planta in co-IP experiments. Among tested constructs, only the Pikp-1:ancHMA_{LVKIE} chimera associated with the effector at levels similar to Pikp-1 (*Figure 4C*, *Figure 4—figure supplement 1*). This indicates that the polymorphic residues in the IAQVV/LVKIE region are critical for the evolution of enhanced AVR-PikD binding in Pikp-1.

## Two substitutions within the IAQVV/LVKIE region of ancHMA increase binding to AVR-PikD

To understand the evolutionary trajectory of the IAQVV/LVKIE region, we set out to reconstruct the evolutionary history of this region. We performed probability-based ASR, based on protein

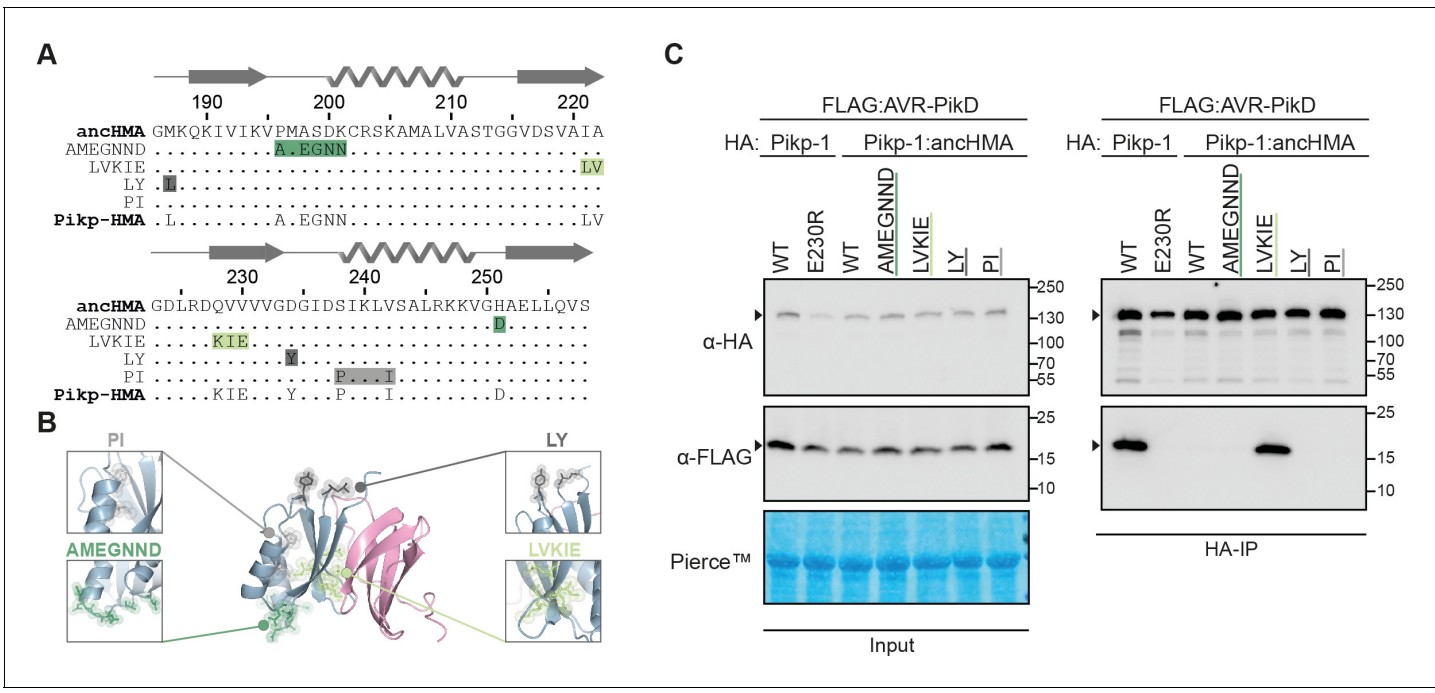

**Figure 4.** The IAQVV/LVKIE region of the Pikp-HMA domain determines high-affinity binding to AVR-PikD. (**A**) Protein sequence alignment showing the Pikp–ancHMA swap chimeras. The amino acid sequences of ancestral HMA (ancHMA), Pikp-HMA, and chimeras are aligned, with the protein model above corresponding to the Pikp-HMA structure. The colour-coded rectangles correspond to polymorphic regions used for chimeric swaps. (**B**) Schematic representation of Pikp-HMA (blue) in complex with AVR-PikD (pink) (*De la Concepcion et al., 2018*), with polymorphic regions between the Pikp-HMA and the ancHMA colour-coded as in (**A**). The molecular surfaces of the polymorphic residues are also shown. (**C**) Association between AVR-PikD (N-terminally tagged with FLAG) and Pikp-1, Pikp-1_{E230R}, Pikp-1:ancHMA, and Pikp-1:ancHMA chimeras (N-terminally tagged with HA), labelled above, was tested in planta in co-IP experiment. Wild-type (WT) Pikp-1 and Pikp-1_{E230R} were used as a positive and negative control, respectively. Immunoprecipitates (HA-IP) obtained with anti-HA probe and total protein extracts (input) were immunoblotted with the appropriate antisera, labelled on the left. Rubisco loading control was performed using Pierce staining solution. Arrowheads indicate expected band sizes. Results from three independent replicates of this experiment are shown in *Figure 4—figure supplement 1*.

The online version of this article includes the following figure supplement(s) for figure 4:

**Figure supplement 1.** Replicates of the co-immunoprecipitation (co-IP) experiment between AVR-PikD and the Pikp-1:ancHMA chimeras.

sequence alignment and a representative phylogeny of 19 K-type integrated HMA domains, where ancHMA was separated from Pikp-HMA by five internal nodes (*Figure 3—figure supplement 2*). Using sequences predicted at these nodes, we identified the three most ancient substitutions at the resolution of single amino acids—I221L, followed by Q228L, followed by V229E (*Figure 5A*). Discerning the order of the two most recent substitutions, Ala-222-Val and Val-230-Glu, was not possible. We generated ancHMA mutants by consecutively introducing historical substitutions into their respective ancestral backgrounds, generating ancHMA$_{LAQVV}$, ancHMA$_{LAKVV}$, and ancHMA$_{LAKIV}$, as well as two plausible alternative states between LAKIV and LVKIE—ancHMA$_{LAKIE}$ and ancHMA$_{LVKIV}$.

To determine the extent to which each of the historical mutations contributed to changes in effector binding, we cloned the ancHMA mutants into the Pikp-1 background and assayed them for AVR-PikD binding in planta. Initial results showed low accumulation levels of Pikp-1:ancHMA$_{LVKIV}$ mutant, preventing meaningful interpretation of results obtained using this protein (*Figure 5—figure supplement 1*), hence, we excluded it from further analysis; the remaining constructs accumulated to similar levels. In co-IP experiments, Pikp-1:ancHMA$_{LVKIE}$ exhibited the strongest association with AVR-PikD followed by Pikp-1:ancHMA$_{LAKIE}$, which displayed intermediate binding (*Figure 5B*, *Figure 5—figure supplement 2*). The remaining mutants did not show gain-of-binding to AVR-PikD when compared to Pikp-1:ancHMA.

To quantify how historical substitutions in the IAQVV/LVKIE region contributed to enhancing AVR-PikD binding, we carried out surface plasmon resonance (SPR) experiments using AVR-PikD and the full set of the ancHMA mutants cloned to match the residues Gly-186–Ser-258 of the full-length Pikp-1, which have previously been successfully used in vitro (*Maqbool et al., 2015*), purified from *Escherichia coli* by a two-step purification method (*Figure 5—figure supplement 3*). We measured binding by monitoring the relative response following AVR-PikD immobilisation on the NTA-sensor chip and injection of the ancHMA proteins at three different concentrations. To capture the binding dynamics, we recorded the response at two timepoints: at the end of HMA injection ('binding') and 15 s post-injection ('dissociation') (*Figure 5—figure supplement 4A*). We normalised the response units to the theoretical maximum response ($R_{max}$) and expressed the results as a percentage of $R_{max}$ (%$R_{max}$), which gave a relative indication of binding strength. Average $\Delta$%$R_{max}$, calculated from a difference between $R_{max}$ for 'binding' and 'dissociation', was used as an off-rate approximate. AncHMA$_{LVKIE}$ formed the strongest interaction with AVR-PikD at levels similar to Pikp-HMA, followed by ancHMA$_{LAKIE}$, then ancHMA$_{LAQVV}$, ancHMA$_{LAKIV}$, and ancHMA, which showed weaker interactions; we did not record any significant binding for ancHMA$_{LAKVV}$ (*Figure 5C*, *Figure 5—figure supplement 4B*; *Supplementary file 1I*). These results indicate that the two most recent mutations, Ala-222-Val and Val-230-Glu, collectively referred to as AV-VE, determined HMA transition towards high-affinity AVR-PikD binding.

We noted from the panel of 19 integrated HMA sequences collected in this study that the AV-VE polymorphisms are unique to Pikp-1 and Pikh-1 of rice. The *Pikp-1* and *Pikh-1* genes are highly similar to each other; out of a total of three polymorphisms, there is only one synonymous substitution that distinguishes their nearly 3500-bp-long coding sequences (*Supplementary file 1J*). Although this precludes a rigorous estimation of evolutionary divergence times of the integrated HMAs, the near-absence of synonymous nucleotide polymorphisms between Pikp-1 and Pikh-1 suggests a very recent emergence of the AV-VE polymorphisms.

## The AV-VE substitutions are sufficient to increase binding affinity towards AVR-PikD

To investigate the role of historical contingency in the evolutionary history of the Pikp-1-integrated HMA domain, we tested the impact of early historical substitutions from the ancestral IAQVV residues to the Pikp-1 LVKIE on effector-binding strength. We bypassed the historical sequence by incorporating the AV-VE mutations directly into ancHMA, generating Pikp:ancHMA$_{IVQVE}$, and examined effector binding in co-IP experiments (*Figure 5—figure supplement 5*). Pikp:ancHMA$_{IVQVE}$ showed stronger association with AVR-PikD than Pikp:ancHMA; however, we were unable to directly compare its association to Pikp:ancHMA$_{LVKIE}$ due to uneven protein accumulation levels. These results indicate that the AV-VE substitutions are sufficient to increase binding affinity towards the AVR-PikD effector independently of the other three polymorphic residues in this IAQVV/LVKIE interface. Nonetheless, we cannot exclude the possibility that prior mutations had quantitative epistatic effects on the interaction that cannot be quantified by co-IP.

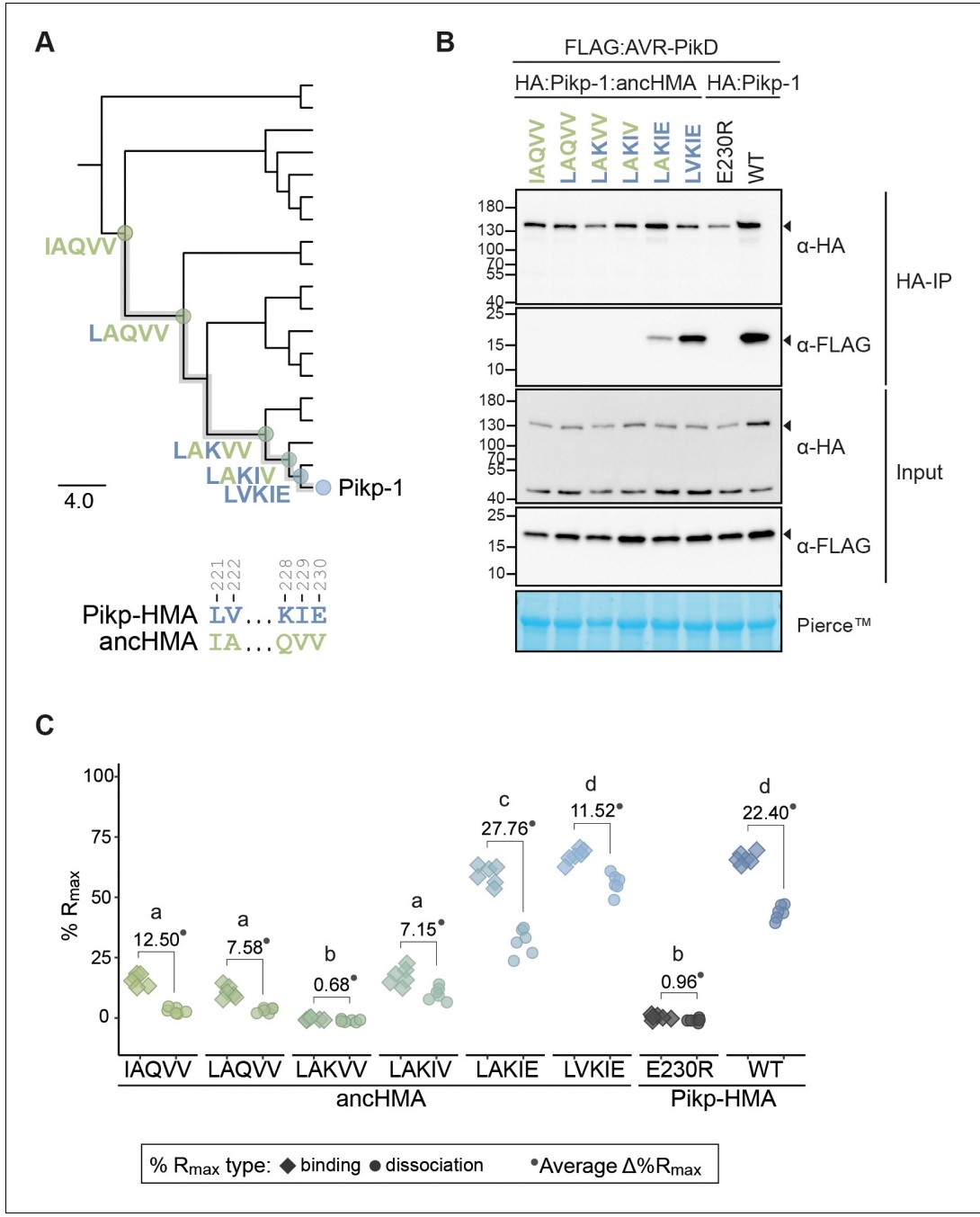

**Figure 5.** The AV-VE substitutions within the IAQVV/LVKIE region of ancestral HMA (ancHMA) increase binding to AVR-PikD. (**A**) Schematic representation of a neighbour joining (NJ) phylogenetic tree of the heavy metal-associated (HMA) domain from *Oryza* spp. (shown in *Figure 3—figure supplement 2*). The scale bar indicates the evolutionary distance based on the number of base substitutions per site. Historical mutations in the IAQVV/LVKIE region acquired over the course of Pikp-HMA evolution are shown next to the appropriate nodes. The mutations are colour-coded to match the ancestral (green) and present-day (blue) states. (**B**) Co-immunoprecipitation (Co-IP) experiment illustrating in planta association of AVR-PikD (N-terminally tagged with FLAG) with Pikp-1 and Pikp-1: ancHMA (N-terminally tagged with HA), labelled above. Wild-type (WT) HA:Pikp-1 and HA:Pikp-1$_{E230R}$ proteins were used as a positive and negative control, respectively. Immunoprecipitates (HA-IP) obtained with anti-HA probe and total protein extracts (input) were immunoblotted with appropriate antibodies (listed on the right). Loading control, featuring Rubisco, was performed using Pierce staining. The arrowheads indicate expected band sizes. Three independent replicates of this experiment are shown in *Figure 5—figure supplement 2*. (**C**) Plot illustrating calculated percentage of the theoretical maximum response (%R$_{max}$) values for interaction of HMA

*Figure 5 continued on next page*

*Figure 5 continued*

analytes, labelled below, with AVR-PikD ligand (featuring C-terminal HIS tag) determined using surface plasmon resonance. %$R_{max}$ was normalised for the amount of ligand immobilised on the NTA-sensor chip. The chart summarises the results obtained for HMA analytes at 400 nM concentration from three independent experiments with two internal repeats. Three different concentrations of the analytes (400 nM, 200 nM, 50 nM) were tested; results for the 200 nM and 50 nM concentrations are shown in *Figure 5—figure supplement 4*. Average Δ%$R_{max}$ (•) values represent absolute differences between values for 'binding' and 'dissociation', calculated from the average values for each sample, and serve as an off-rate approximate. Statistical differences among the samples were analysed with Tukey's honest significant difference (HSD) test (p<0.01); p-values for all pairwise comparisons are presented in *Supplementary file 1I*.

The online version of this article includes the following source data and figure supplement(s) for figure 5:

**Source data 1.** Raw data of Pikp-ancHMA $R_{max}$ SPR.

**Figure supplement 1.** Co-immunoprecipitation experiment between AVR-PikD and the two plausible historical states of the IAQVV/LVKIE region within Pikp-HMA.

**Figure supplement 2.** Replicates of the co-immunoprecipitation (co-IP) experiments between the Pikp-1:ancHMA IAQVV/LVKIE mutants and AVR-PikD.

**Figure supplement 3.** Purified proteins used in surface plasmon resonance studies.

**Figure supplement 4.** Surface plasmon resonance (SPR) results show the effect of the IAQVV-LVKIE mutations on the AVR-PikD binding, as indicated by %$R_{max}$.

**Figure supplement 5.** The AV-VE (Ala-222-Val and Val-230-Glu) substitutions are sufficient to increase binding affinity towards the AVR-PikD effector in co-immunoprecipitation (co-IP).

## High binding affinity to AVR-PikD accounts for the capacity of Pikp-1: ancHMA to trigger an immune response

To test if effector binding by Pikp-1:ancHMA is sufficient to trigger an immune response, we performed HR cell death assays by transiently co-expressing each of the Pikp-1:ancHMA fusions with AVR-PikD and Pikp-2 in *Nicotiana benthamiana*. We discovered that all Pikp-1:ancHMA variants are autoactive and trigger spontaneous cell death in the absence of the effector (*Figure 6—figure supplement 1*, *Figure 6—figure supplement 2*). Notably, the presence of the Pikp-2 partner is required for Pikp-1:ancHMA autoactivity.

Next, we used previously generated ancHMA chimeras to delimitate the region responsible for the autoactivity phenotype of Pikp-1:ancHMA. We tested these fusions for loss of function in cell death assays by transient co-expression with Pikp-2 in *N. benthamiana* (*Figure 6—figure supplement 3*, *Figure 6—figure supplement 4*). Among these, Pikp-1:ancHMA$_{AMEGNND}$ was the only chimera to show complete loss of autoactivity. This phenotype was not due to protein instability or low protein abundance (*Figure 4C*, *Figure 4—figure supplement 1*). These results suggest that the PMASDKH/AMEGNND region, located in the β1–α1 and α2–β4 loops of the Pikp-HMA domain, underpins Pikp-1:ancHMA autoactivity.

To determine whether gain of AVR-PikD binding results in a functional immune response, we performed cell death assays using Pikp-1:ancHMA mutants in the IAQVV/LVKIE region. We first removed autoactivity by introducing AMEGNND mutations into these constructs (*Figure 6A*), henceforth called Pikp-1:ancHMA$_{LVKIE}$*, Pikp-1:ancHMA$_{LAKIE}$*, Pikp-1:ancHMA$_{LAKIV}$*, Pikp-1:ancHMA$_{LAKVV}$*, Pikp-1:ancHMA$_{LAQVV}$*. None of the resulting mutants triggered spontaneous cell death when transiently co-expressed with Pikp-2 (*Figure 6B*, C *Figure 6—figure supplement 5*). Co-expression with AVR-PikD revealed that the strength of binding directly correlates with the strength of HR. The mutants that gained AVR-PikD binding in the co-IP and SPR experiments, namely Pikp-1:ancHMA$_{LAKIE}$* and Pikp-1:ancHMA$_{LVKIE}$*, showed HR phenotypes. The Pikp-1:ancHMA$_{LVKIE}$* mutants triggered cell death at levels similar to Pikp-1, whereas the HR triggered by Pikp-1:ancHMA$_{LAKIE}$* was slightly, yet significantly, reduced when compared to Pikp-1. By contrast, Pikp-1:ancHMA*, Pikp-1:ancHMA$_{LAKVV}$*, and Pikp-1:ancHMA$_{LAQVV}$* did not elicit cell death above background levels. All proteins accumulated at similar levels in western blot analysis (*Figure 6—figure supplement 6*). Overall, these results indicate that the adaptive mutations in the IAQVV/LVKIE region towards AVR-PikD binding at high affinity also enable effector-dependent activation of the cell death immune response.

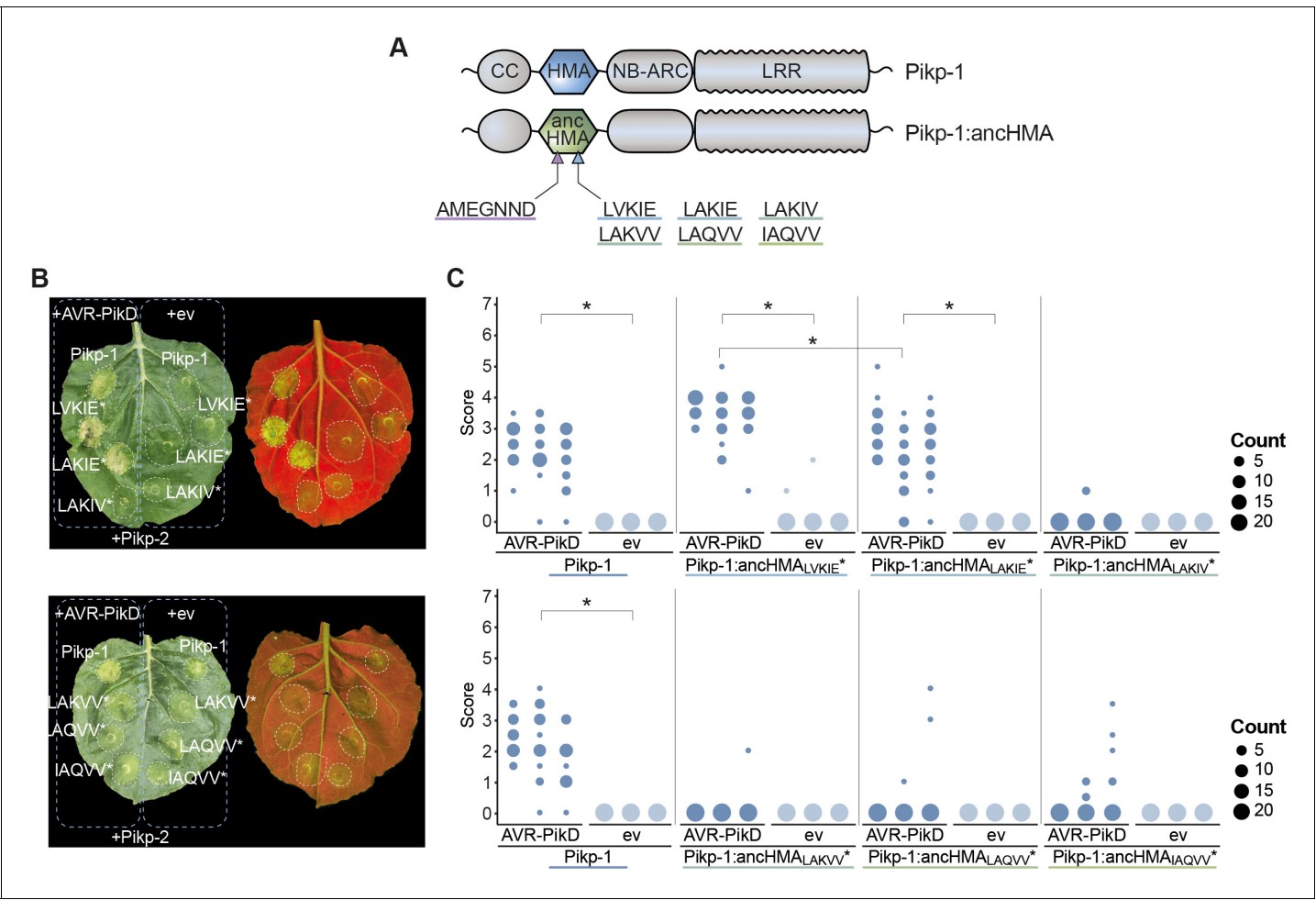

**Figure 6.** Pikp-1:ancHMA_LVKIE* and Pikp-1:ancHMA_LAKIE* mediate immune response towards the AVR-PikD effector. (**A**) Schematic representation of wild-type Pikp-1 and Pikp-1:ancHMA fusions used in the assay. The mutated regions are presented with arrowheads and listed. (**B**) Representative images of hypersensitive response (HR) cell death assay after transient co-expression of the Pikp-1:ancHMA* mutants (C-terminally tagged with HF) with AVR-PikD (N-terminally tagged with Myc) and Pikp-2 (C-terminally tagged with HA). Empty vector (ev) was used as a negative control. All constructs were co-expressed with the gene silencing suppressor p19 (**Win and Kamoun, 2003**). The leaves were photographed 5 days after infiltration under daylight (left) and UV light (right). (**C**) HR was scored at 5 days post-agroinfiltration. The results are presented as dot plots, where the size of a dot is proportional to the number of samples with the same score (count) within the same biological replicate. The experiment was independently repeated at least three times with 23–24 internal replicates; the columns within tested conditions (labelled on the bottom) correspond to results from different biological replicates. Significant differences between relevant conditions are marked with an asterisk (*); details of the statistical analysis are summarised in **Figure 6—figure supplement 5**.

The online version of this article includes the following source data and figure supplement(s) for figure 6:

**Source data 1.** Hypersensitive response scores for IAQVV to LVKIE mutations in Pikp-HMA.

**Figure supplement 1.** Pikp-1:ancHMA fusions are autoactive in a Pikp-2-dependent manner.

**Figure supplement 1—source data 1.** Hypersensitive response scores used in **Figure 6—figure supplement 1**.

**Figure supplement 2.** Statistical analysis of hypersensitive response cell death for the Pikp-1:ancHMA fusions.

**Figure supplement 3.** The AMEGNND mutations within ancestral HMA (ancHMA) abolish autoactivity.

**Figure supplement 3—source data 1.** Hypersensitive response scores used in Figure6—figure supplement 3.

**Figure supplement 4.** Statistical analysis of cell death assay for the Pikp-1:ancHMA chimeras.

**Figure supplement 5.** Statistical analysis of cell death for the Pikp-1:ancHMA mutants within the IAQVV/LVKIE region.

**Figure supplement 6.** In planta accumulation of the Pikp-1:ancHMA* mutants in the IAQVV/LVKIE region.

## A distinct region (MKANK/EMVKE) in the integrated HMA domain of Pikm-1 determines high-affinity AVR-PikD binding

As noted above, the LVKIE polymorphisms are relatively rare among Pik-1 allelic variants and *Oryza* orthologues (2 out of 19 examined sequences) (*Figure 7—figure supplement 1*). Other rice allelic variants of Pik-1 retain the predicted IAQVV ancestral state. Interestingly, Pikm-1, a Pik-1 allelic variant with the IAQVV residues, binds the AVR-PikD effector with high affinity and triggers an immune response upon effector recognition (*De la Concepcion et al., 2018*; *Kanzaki et al., 2012*). This led us to hypothesise that the integrated HMA domain of Pikm-1 (Pikm-HMA) has undergone a distinct evolutionary path towards AVR-PikD binding compared to Pikp-HMA.

To determine which Pikm-HMA mutations have enabled gain of AVR-PikD binding, we performed structure-informed sequence comparison of the Pikm-HMA and ancHMA domains similar to the approach described above for Pikp-1. We amended the sequence of previously predicted ancHMA with a three-amino-acid-long extension (residues 262–264 of the full-length Pikm-1) that includes residues that are polymorphic in Pikm-HMA but identical between ancHMA and Pikp-HMA. Next, we mapped five polymorphic regions that differentiate the ancHMA from modern Pikm-HMA (*Figure 7A, B*), introduced mutations in these regions in Pikm-1:ancHMA, and subjected the Pikm-1:ancHMA variants to in planta co-IP with AVR-PikD. Among the five chimeras tested in this experiment, Pikm-1:ancHMA$_{EMVKE}$ was the only one to associate with AVR-PikD (*Figure 7C*, *Figure 7—figure supplement 2*). Among the remaining chimeras, Pikm-1:ancHMA$_{VH}$ protein was unstable and hence yielded inconclusive results. Overall, we conclude that Pikm-HMA evolved towards association with AVR-PikD through mutations in the MKANK/EMVKE region, a distinct interface from the IAQVV/LVKIE region of Pikp-1.

## The ANK-VKE mutations confer high-affinity AVR-PikD binding in Pikm-HMA

We reconstructed the mutational history of the MKANK/EMVKE interface to trace the evolutionary trajectory of Pikm-HMA detection of AVR-PikD (*Figure 8A*). The ASR was performed by a combination of manual and probability-based approaches using a protein sequence alignment and a representative phylogenetic tree of the HMA domain, where ancHMA and Pikm-HMA were separated by four internal nodes (*Figure 3—figure supplement 2*). However, we could only identify one node that represents an evolutionary intermediate between the ancestral MKANK and present-day EMVKE states, namely EMANK, that emerged through MK-EM mutations (M188E and K189M). The ANK-VKE mutations (A261V, N262K, and K263E) were acquired at a later timepoint, and determining the order of individual mutations was not possible given the limits of the phylogenetic tree resolution.

To evaluate the impact of these historical mutations, we generated the ancHMA$_{EMANK}$ mutant that recapitulates the predicted step-by-step intermediate state of the MKANK/EMVKE region, incorporated this mutant into the Pikm-1 backbone, and assayed it for in planta association with AVR-PikD. By contrast to Pikm:ancHMA$_{EMVKE}$, Pikm:ancHMA$_{EMANK}$ did not gain the capacity to associate with AVR-PikD relative to Pikm:ancHMA$_{MKANK}$ (*Figure 8B*, *Figure 8—figure supplement 1*).

Next, we validated these results in vitro using the AVR-PikD protein and the full set of ancHMA mutants purified from *E. coli* (*Figure 8—figure supplement 2*). To encompass the full diversity between the ancestral and present-day states of Pikm-HMA, we used HMA sequences with a five-amino acid extension at the C-terminus (ancHMA+5) compared to the constructs used in the Pikp-HMA experiments. During protein purification, we noted a shift in elution volume of the ancHMA+5 in complex with AVR-PikD relative to the elution volume of the ancHMA$_{LVKIE}$–AVR-PikD complex in size-exclusion chromatography (*Figure 8—figure supplement 3*). We concluded that this shift is consistent with different stoichiometries of the ancHMA–AVR-PikD complexes; while ancHMA$_{LVKIE}$–AVR-PikD formed a two-to-one complex, the constructs with the extension interacted with the effector at a one-to-one ratio. Accounting for this stoichiometry, we carried out SPR experiments using the same experimental design as in the Pikp-HMA assays and discovered that among tested mutants the ancHMA$_{EMVKE}$ displayed the highest rates of interaction with AVR-PikD, followed by ancHMA$_{EMANK}$ and ancHMA$_{MKANK}$. Although we noted that all tested HMA mutants exhibited similar binding affinity to AVR-PikD at 400 nM concentration (*Figure 8—figure supplement 4*; *Supplementary file 1L*), they displayed marked differences in the shapes of their sensorgrams (*Figure 8C, D*, *Figure 8—figure supplement 4*, *Figure 8—figure supplement 5*). First, despite high values for 'binding',

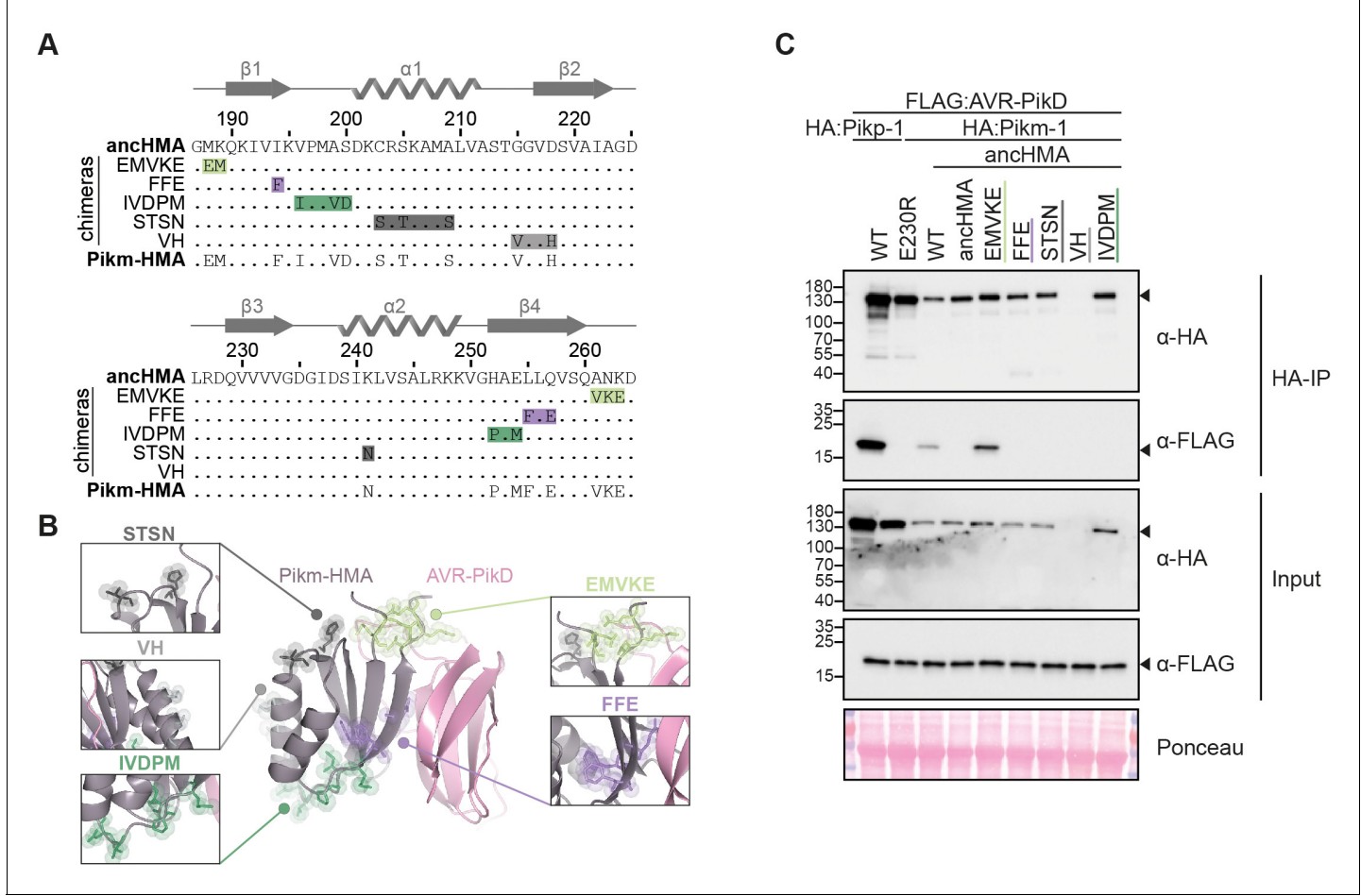

**Figure 7.** The MKANK/EMVKE region of the heavy metal-associated (HMA) domain of Pikm-1 determines high-affinity AVR-PikD binding. (**A**) Protein sequence alignment between the ancestral HMA (ancHMA), Pikm-HMA, and Pikm–ancHMA chimeras. The protein model above the alignment depicts Pikm-HMA secondary structure. The colour-coded rectangles mark polymorphic regions used for chimeric swaps. (**B**) Schematic representation of the Pikm-HMA domain (purple) in complex with AVR-PikD (pink) (*De la Concepcion et al., 2018*), with polymorphic regions between Pikm-HMA and ancHMA colour-coded as in (**A**). The molecular surfaces of the polymorphic residues are also shown. (**C**) EMVKE substitutions in the ancHMA restore in planta association with AVR-PikD. Co-immunoprecipitation experiment between AVR-PikD (N-terminally tagged with FLAG) and Pikp-1:ancHMA chimeras (N-terminally tagged with FLAG), labelled above. Wild-type (WT) Pikp-1/Pikm-1 and Pikp-1$_{E230R}$ were used as positive and negative controls, respectively. Immunoprecipitates (HA-IP) obtained with anti-HA probe and total protein extracts (input) were immunoblotted with the appropriate antisera (labelled on the right). Rubisco loading control was carried out using Ponceau staining. Arrowheads indicate expected band sizes. Three independent replicates of this experiment are shown in *Figure 7—figure supplement 2*.

The online version of this article includes the following figure supplement(s) for figure 7:

**Figure supplement 1.** Protein sequence alignment of the heavy metal-associated (HMA) domain from the *Oryza* spp.

**Figure supplement 2.** Replicates of the co-immunoprecipitation (co-IP) experiment between the Pikm-1:ancHMA chimeras and AVR-PikD.

ancHMA exhibited high off-rates, as illustrated by the pattern of 'dissociation' and shape of the curves. Second, ancHMA$_{EMVKE}$ displayed high values for 'binding' and 'dissociation', with low $\Delta\%$ R$_{max}$, indicating tight and stable binding. Finally, ancHMA$_{EMANK}$ fell in-between ancHMA and ancHMA$_{EMVKE}$, with stable and relatively low $\Delta\%$R$_{max}$ at the top concentration and moderate $\Delta\%$ R$_{max}$ at lower concentrations. These findings indicate that the ANK-VKE substitutions are essential for Pikm-HMA high-affinity binding of AVR-PikD. Altogether, both co-IP and SPR experiments indicate that the MKANK/EMVKE region plays an important role in high-affinity binding of the AVR-PikD effector by Pikm-HMA.

We further noted that the ANK-VKE substitutions are present in three Pik-1 alleles of rice, namely closely related Pik*-1 (*Zhai et al., 2011*), Pikm-1 (*Ashikawa et al., 2008*), and Piks-1 (*Jia et al., 2009*; *Figure 7—figure supplement 1*). Pikm-1 differs from Piks-1 and Pik*-1 by only two and eight

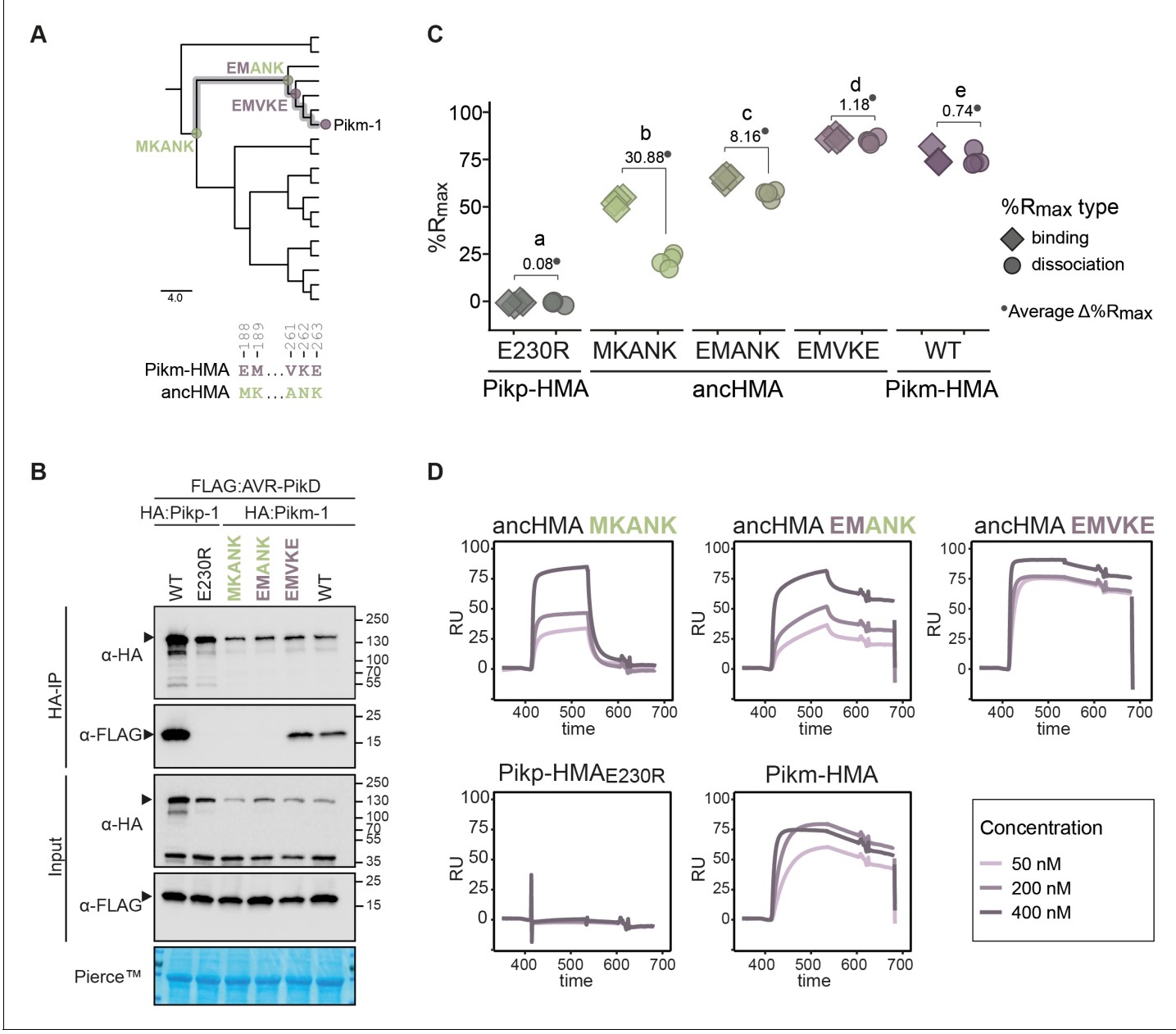

**Figure 8.** The ANK-VKE substitutions are essential for Pikm-HMA adaptation towards high-affinity binding to AVR-PikD. (**A**) Schematic representation of the neighbour joining (NJ) tree of the ancestral HMA (HMA) domains from *Oryza* spp. (shown in *Figure 3—figure supplement 2*). The scale bar indicates the evolutionary distance based on the number of base substitutions per site. Historical substitutions in the MKANK/EMVKE region acquired over the course of Pikm-HMA evolution are shown next to the corresponding nodes. The mutations are colour-coded to match the ancestral (green) and present-day (purple) states. (**B**) Co-immunoprecipitation experiment illustrating in planta association of AVR-PikD (N-terminally tagged with FLAG) with Pikm-1 and Pikm-1:ancHMA proteins (N-terminally tagged with HA), labelled above. Wild-type (WT) Pikp-1/Pikm-1 and Pikp-1$_{E230R}$ constructs were used as positive and negative controls, respectively. Immunoprecipitates (HA-IP) obtained using anti-HA probes and total protein extracts (input) were immunoblotted with the appropriate antisera (depicted on the left). The arrowheads indicate expected band sizes. Rubisco loading control was performed using Pierce solution. Three independent replicates of this experiment are shown in *Figure 8—figure supplement 1*. (**C**) Plot illustrating calculated percentage of the theoretical maximum response (%R$_{max}$) values for interaction of heavy metal-associated (HMA) analytes, labelled below, with AVR-PikD ligand (C-terminally tagged with HIS) determined by surface plasmon resonance (SPR). %R$_{max}$ was calculated assuming a one-to-one (HMA-to-effector) binding model for Pikm-HMA and ancHMAs, and a two-to-one for Pikp-1$_{E230R}$. The values were normalised for the amount of ligand immobilised on the NTA-chip. The chart summarises the results obtained for HMA analytes at 200 nM concentration from five independent experiments, with all the data points represented as diamonds ('binding') or circles ('dissociation'). Three different concentrations of analytes (400 nM, 200 nM, 50 nM) were tested; results for 400 nM and 50 nM concentrations are shown in *Figure 8—figure supplement 4*. Average Δ%R$_{max}$ (•) values represent absolute differences between values for 'binding' and 'dissociation', calculated from the average values for each sample, and serve as an off-

*Figure 8 continued on next page*

*Figure 8 continued*

rate approximate. Statistical differences among the samples were analysed with Tukey's honest significant difference (HSD) test (p<0.01); p-values for all pairwise comparisons are presented in **Supplementary file 1K**. (D) The SPR sensorgrams of the AVR-PikD and HMA proteins, corresponding to the data used in (C). Independent replicates of this experiment are presented in **Figure 8—figure supplement 5**.

The online version of this article includes the following source data and figure supplement(s) for figure 8:

**Source data 1.** Raw data of Pikm-ancHMA $R_{max}$ SPR.
**Figure supplement 1.** Replicates of the co-immunoprecipitation experiment between Pikm-1:ancHMA mutants in the MKANK/EMVKE region and AVR-PikD.
**Figure supplement 2.** Purified proteins used in surface plasmon resonance studies.
**Figure supplement 3.** Different stoichiometry of the ancHMA–AVR-PikD complexes.
**Figure supplement 4.** Surface plasmon resonance (SPR) results showing the effect of the step-by-step mutations within the MKANK/EMVKE region on the AVR-PikD binding in vitro, as indicated by %$R_{max}$.
**Figure supplement 5.** The surface plasmon resonance (SPR) sensorgrams for the AVR-PikD–HMA binding.

amino acid polymorphisms, respectively, but no synonymous changes (**Supplementary file 1J**). This demonstrates a very recent emergence of these Pik-1 alleles and their associated ANK-VKE substitutions. Next, we aimed to determine whether the gain of AVR-PikD binding translates to an immunoactive Pikm-1:ancHMA by means of HR cell death assay. However, while addressing this question we run into several technical problems, including (1) autoactivity of Pikm-1:ancHMA, (2) perturbed response to AVR-PikD, (3) reduced protein accumulation levels, and (4) weak/inconsistent HR (**Białas et al., 2021**). These precluded reliable studying of how the strength of AVR-PikD binding correlates with HR cell death.

## Pikp-1 and Pikm-1 NLR receptors convergently evolved through distinct biochemical paths to gain high-affinity AVR-PikD binding

Our findings led us to develop an evolutionary model that depicts convergent molecular evolution of Pikp-1 and Pikm-1 towards AVR-PikD binding (**Figure 9**). To interpret this model from a structural perspective, we attempted to determine crystal structures of the ancHMA domains in complexes with AVR-PikD. Crystallisation screens of the heterologously expressed proteins resulted in crystals of the ancHMA_LVKIE–AVR-PikD complex, which diffracted to 1.32 Å resolution (**Supplementary file 1L**). The structure revealed an overall architecture of the complex similar to that of previously published co-structures of Pik-HMAs and AVR-PikD (**Figure 9—figure supplement 1A**; **De la Concepcion et al., 2018**; **De la Concepcion et al., 2021**; **Maqbool et al., 2015**). We note that the MKANK/ EMVKE and IAQVV/LVKIE regions map to two of the three interaction interfaces previously described to underpin binding of AVR-PikD, and other AVR-Pik variants, to Pik-HMAs (**De la Concepcion et al., 2021**; **De la Concepcion et al., 2019**; **De la Concepcion et al., 2018**).

To gain insights into the structural determinants of effector binding in the IAQVV/LVKIE region, we generated a homology model of the ancHMA in complex with AVR-PikD (**Figure 9—figure supplement 1B**). We further validated modelled interactions by examining the published structure of Pikm-HMA (**De la Concepcion et al., 2018**), whose IAQVV/LVKIE region is identical to ancHMA. Close inspection of these structures revealed that the Val-230-Glu (V230E) substitution enhances the interaction with AVR-PikD through hydrogen bond formation with His-46 (**Figure 9A**, **Figure 9—figure supplement 1C**). This bond is formed by Glu-230 (E-230) of ancHMA_LVKIE but absent in Pikm-HMA and ancHMA, which carry Val-230 (V-230) at the structurally equivalent position.

Next, we examined the structural basis of the interaction of the MKANK/EMVKE region with AVR-PikD by comparing Pikm- and Pikp-HMA structures (**De la Concepcion et al., 2018**) that feature EMVKE and LKANK residues (reminiscent of the MKANK amino acids present in ancHMA), respectively. In both cases, Lys-262 (K262) is a major effector-binding determinant that forms hydrogen bonds or salt bridges with Glu-53 and Ser-72 of AVR-PikD (**Figure 9A**). However, in Pikm-HMA the position of Lys-262 (K262) is structurally shifted causing a difference in the conformation of the HMA peptide backbone, and associated side chains, compared to Pikp-HMA. Homology modelling fails to predict this change in the HMA backbone that results in tighter interaction between AVR-PikD and Pikm-HMA compared to Pikp-HMA (**De la Concepcion et al., 2021**; **De la Concepcion et al., 2019**; **De la Concepcion et al., 2018**). We conclude that Asn-262-Lys (N262K) and Lys-263-Glu (K263E) of

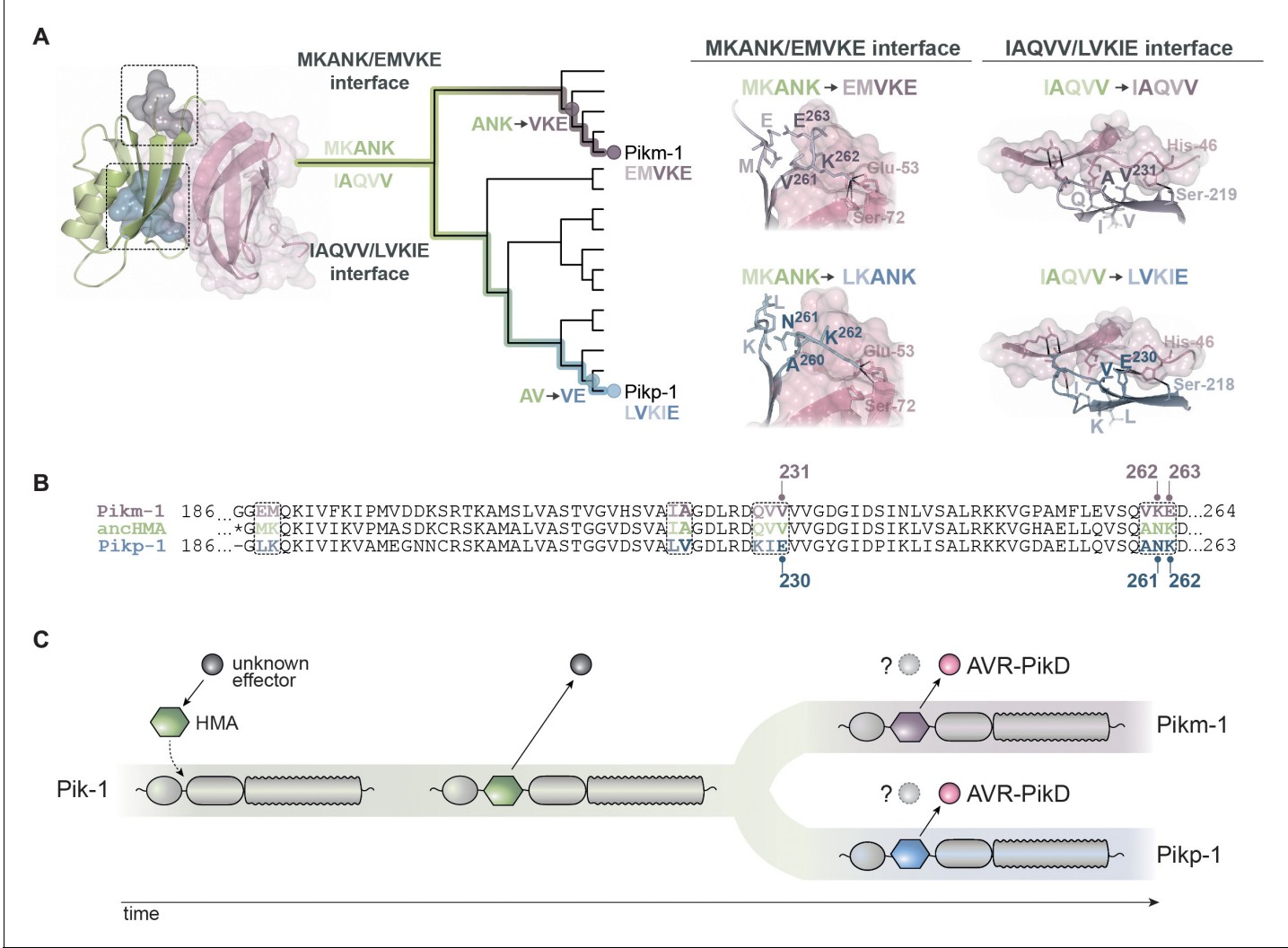

**Figure 9.** Model of molecular convergence of Pikp-1 and Pikm-1 towards AVR-PikD binding at high affinity. (**A**) The heavy metal-associated (HMA) domains of Pikp-1 and Pikm-1 receptors have convergently evolved through distinct evolutionary and biochemical paths to bind AVR-PikD with high affinity. The Pikp-HMA domain evolved through the AV-VE adaptations in the IAQVV/LVKIE region, whereas Pikm-HMA domain acquired the ANK-VKE mutations in the MKANK/EMVKE region. Schematic representations of the HMA–AVR-PikD structures, adapted from *De la Concepcion et al., 2018*, are presented with selected side chains shown as sticks and labelled; the colours of the residue labels match colours of the respective molecules. Dashed lines stand for hydrogen bonds or salt bridges. (**B**) The protein sequence alignment between Pikp-HMA, Pikm-HMA, and ancestral HMA (ancHMA), with relevant amino acids marked. (**C**) We propose a model in which the HMA effector target integrated into Pik-1 to bait the recognition of an unknown effector. Throughout evolution the Pik-1 receptor and its integrated HMA domain diversified and led to the emergence of the Pikp-1 and Pikm-1 allelic variants that bind newly emerged AVR-PikD effector.

The online version of this article includes the following figure supplement(s) for figure 9:

**Figure supplement 1.** The Val-230-Glu mutation within the LVKIE region of ancestral HMA (ancHMA) enhances interaction with AVR-PikD through hydrogen bond formation.

the ANK-VKE substitution likely determine differential binding between the ancestral and present-day Pikm-HMA domains.

## Discussion

The molecular evolution events associated with the transition of NLR IDs from pathogen effector targets to baits remain elusive. Here, we investigated the evolution of these unconventional domains of NLR receptors using rice Pik as a model system. First, we performed extensive phylogenetic analyses

to determine that the integration of the HMA domain emerged over 15 MYA, predating the radiation of Oryzinae (*Figure 1D*). Using sequence reconstruction and resurrection of an ancestral integrated HMA domain that dates back to early divergence of *Oryza* spp., we showed that the capacity of Pik-1 to sense and respond to AVR-PikD evolved relatively recently through distinct evolutionary and biochemical paths in two alleles of Pik-1, Pikp-1 and Pikm-1. This combination of evolutionary and biochemical approaches allowed us to develop a model of the adaptive evolution of the Pik proteins towards high-affinity AVR-PikD binding (*Figure 9*).

The molecular bases of functional transitions in NLR evolution remain poorly understood, especially over extended timescales. Here, we showed that adaptive evolution of Pikp-1 and Pikm-1 from weak to high-affinity binding to the AVR-PikD effector involves two distinct regions within the HMA domain. Overall, these interfaces seem to function in a synergistic yet interchangeable manner, such that weak interaction at one interface can be compensated by strong interaction at a different one (*De la Concepcion et al., 2021*; *De la Concepcion et al., 2018*). We propose that this modularity between different regions of the HMA increases the HMA's capacity for rapid adaptive evolution as it can follow alternative mutational paths to produce similar phenotypic outcomes and counteract rapidly evolving pathogen effectors. Indeed, HMA domains can also detect another *M. oryzae* effector AVR-Pia through an alternative interface (*Guo et al., 2018*; *Varden et al., 2019*), further illustrating the capacity of the HMA domain to bait pathogen effectors through different interfaces. This may have contributed to the recurrent emergence of HMAs as NLR IDs. Previous studies have revealed that HMAs have independently integrated into NLR immune receptors from at least four flowering plant families (*Kroj et al., 2016*; *Sarris et al., 2016*).

The HMA domain of Pik-1 exhibits signatures of positive selection in contrast to the NB-ARC domain (*Figure 2*), likely reflecting coevolution with pathogen effectors versus overall purifying selection. This further suggests that HMA domains are malleable platforms that can accommodate accelerated mutational rates (*Białas et al., 2018*; *Costanzo and Jia, 2010*). Similar observations have previously been made in a number of plant NLRs, whose individual domains display patterns of asymmetrical evolution and distinct rates of selection, suggesting that NLRs evolve in a modular fashion (*Kuang et al., 2004*; *Maekawa et al., 2019*; *Prigozhin and Krasileva, 2020*; *Read et al., 2020*; *Seeholzer et al., 2010*). Moreover, having a domain responsible for effector recognition may release other domains from the pressure of diversification and reduce the risk of compromising or mis-regulating NLR activity (*Cesari, 2018*). In addition, coupling with a helper NLR such as Pik-2 likely provides yet another mechanism of functional compartmentalisation, further enhancing the evolvability of the sensor by freeing it from the constraint of executing the hypersensitive cell death (*Adachi et al., 2019*; *Cesari, 2018*; *Wu et al., 2018*).

We showed that the evolutionarily derived AV-VE in Pikp-1 (*Figure 5*) and ANK-VKE polymorphisms in Pikm-1 (*Figure 8*) enabled high-affinity binding to AVR-PikD. Although the high sequence divergence and elevated mutation rates among HMA sequences precluded rigorous dating of the emergence of these key adaptations, the low level of total nucleotide polymorphisms among closely related Pik alleles—in particular, the very few synonymous substitutions among Pikp- and Pikm-related alleles—points to a very recent emergence of the adaptive polymorphisms. Given that the rice-infecting lineage of *M. oryzae* is estimated to have arisen about 7000–9000 years ago (*Couch et al., 2005*; *Latorre et al., 2020*), our findings are consistent with the view that Pik-1 alleles evolved during rice domestication as previously suggested (*Kanzaki et al., 2012*; *Zhai et al., 2011*). In addition, AVR-Pik is widespread in rice-infecting isolates but absent in other blast fungus lineages (*Bentham et al., 2021*; *Langner et al., 2021*; *Latorre et al., 2020*; *Yoshida et al., 2016*). Therefore, it is tempting to speculate that the rice agroecosystem has created the ecological context that led to Pik neofunctionalisation towards recognition of the new pathogen threat imposed by the blast fungus. Different rice populations may have independently encountered fungal pathogens carrying AVR-Pik, leading to intense natural selection and independent emergence of the Pikp and Pikm adaptations.

We concluded that the Pik-1-integrated HMA domain did not function in sensing AVR-PikD for most of its over 15-million-year-long evolutionary history, inviting the question about the role of the ancestral integrated HMA. It is likely that over millions of years, prior to rice domestication, the Pik-1 HMA domain had recognised effectors other than AVR-Pik. These could be other members of AVR-Pik–like (APikL) effector family (*Bentham et al., 2021*) or their ancestors, the structurally related MAX-effectors—an ancient effector family present across blast lineages and other fungal pathogens

(*de Guillen et al., 2015*; *Petit-Houdenot et al., 2020*)—or effectors from other plant pathogen taxa. Indeed, the HMA domain is known to bind effectors from diverse pathogens including bacteria and oomycetes, in addition to fungi (*González-Fuente et al., 2020*). *Karasov et al., 2014* proposed that NLRs caught in pairwise arms races (one NLR recognising one effector) are likely to be short-lived, whereas NLRs entangled in diffuse evolution (functioning against multiple effectors and/or multiple pathogens) are more likely to persist over longer timescales. Our model paints a more complex picture of the macroevolutionary dynamics of NLR-IDs. These receptors have the capacity to switch from one effector to another, while also engaging in short-term arms race dynamics, as seems to be the case of Pik-1 vs. AVR-Pik (*Białas et al., 2018*; *Kanzaki et al., 2012*). It is remarkable that the *Pik-1* gene and its paired *Pik-2* gene have been maintained in grass populations for tens of millions of years, even after the integration of the HMA domain. This points to a successful evolutionary strategy for generating long-lived disease resistance traits, with HMA promiscuity towards pathogen effectors at the centre of this model.

We discovered that the Pikp-1:ancHMA fusions trigger spontaneous hypersensitive cell death when co-expressed with Pikp-2 and mapped the region responsible for the autoactivity to two HMA parallel loops, β1–α1 and α2–β4 (*Figure 6—figure supplement 1*, *Figure 6—figure supplement 3*). Although the precise mechanism underpinning this autoactivity remains to be elucidated, we propose that coevolution of the HMA with the canonical domains of Pik-1 and/or Pik-2 drives this molecular incompatibility. Mismatching domains from different evolutionary timepoints may disrupt fine-tuned biochemical interactions between HMA and other domains. Indeed, intra- and intermolecular incompatibilities of NLRs are known causes of autoimmunity in plants (*Harris et al., 2013*; *Li et al., 2020b*; *Lukasik-Shreepaathy et al., 2012*; *Qi et al., 2012*; *Rairdan and Moffett, 2006*; *Tran et al., 2017*; *Wang et al., 2015*). We further noted that some Pik-1 orthologues, namely *LpPik-1* and N-type *Pik-1* genes, carry large deletions within their HMAs, which may have emerged to eliminate autoimmunity (*Figure 2—figure supplement 1*). This is consistent with the view that the risk of autoactivity acts as a strong evolutionary constraint narrowing NLR mutational pathways (*Chae et al., 2014*).

We uncovered a rich genetic diversity of *Pik* genes beyond *Oryza* species (*Mizuno et al., 2020*; *Stein et al., 2018*; *Zhai et al., 2011*; *Figure 1*). This enabled us to date the emergence of the *Pik* pair to before the split of two major grass lineages: the BOP and PACMAD clades, which corresponds to 100–50 MYA (*Hodkinson, 2018*). Furthermore, we estimated that Pik-1 acquired the HMA domain prior to the emergence of Oryzinae but after the split from Panicoideae, between 15 and 100 MYA (*Hodkinson, 2018*; *Jacquemin et al., 2011*; *Stein et al., 2018*). Remarkably, the vast majority of Pik-2 and Pik-1 orthologues across the Poaceae exist as genetically linked pairs in a head-to-head orientation. This applies to Pik-1 orthologues with and without the HMA domain, indicating that Pik-1 and Pik-2 pairing occurred prior to HMA integration. Tight genetic linkage of paired NLRs, such as Pik-1/Pik-2 (*Ashikawa et al., 2008*), RGA5/RGA4 (*Cesari et al., 2013*; *Okuyama et al., 2011*), RRS1/RPS4 (*Saucet et al., 2015*), or RPP2A/RPP2B (*Sinapidou et al., 2004*), is thought to facilitate coregulation and coevolution, thereby ensuring proper cooperation between these NLRs and reducing the genetic load caused by autoimmunity (*Baggs et al., 2017*; *Griebel et al., 2014*; *Wu et al., 2018*). However, *Pik-1* and *Pik-2* paralogues also occur adjacent to the paired genes—a phenomenon previously observed in wild and cultivated rice (*Mizuno et al., 2020*)—raising the possibility that these *Pik* genes may form an NLR receptor network beyond the Pik-1/Pik-2 pair (*Wu et al., 2018*). In the future, it would be interesting to investigate the functions of paired Pik-1/Pik-2 and their paralogues and determine whether functional pairing and genetic linkage with *Pik-2* predisposed *Pik-1* for the HMA integration.

In summary, our study illustrates the value of ASR—a method that has rarely been used in the field of plant–microbe interactions (*Dong et al., 2014*; *Tanaka et al., 2019*; *Zess et al., 2019*)—in transcending phylogenetic inference to yield a more elaborate evolutionary model. ASR combined with biochemical and biophysical studies enabled us to determine the directionality of evolution and therefore develop an experimentally validated model of NLR adaptation. The Pik-1/Pik-2 receptor pair emerged as an excellent system to not only provide a framework for drawing links between NLR structure and function but also to place this knowledge in an evolutionary context. This adds to our understanding of selection forces, historical contingency, and functional constraints shaping NLR activities. This approach illustrates how mechanistic research structured by a robust evolutionary framework can enhance our understanding of plant–microbe systems.

# Materials and methods

## Key resources table

| Reagent type (species) or resource | Designation | Source or reference | Identifiers | Additional information |
|---|---|---|---|---|
| Recombinant DNA reagent | pICH41308 | Addgene | No. 47998 | Golden Gate level 0 acceptor |
| Recombinant DNA reagent | pICSL12008 | TSL (The Sainsbury Laboratory) SynBio team | | 35S + Ω promoter Golden Gate module |
| Recombinant DNA reagent | pICH41414 | Addgene | No. 50337 | 35S terminator Golden Gate module |
| Recombinant DNA reagent | pICSL30007 | TSL (The Sainsbury Laboratory) SynBio team | | N-terminal 6×HA Golden Gate module |
| Recombinant DNA reagent | pICH47732 | Addgene | No. 48001 | Level 1 binary vector |
| Recombinant DNA reagent | p41308-PikpN | This paper | | Materials and methods: Cloning for in planta assays |
| Recombinant DNA reagent | p41308-PikpC | This paper | | Materials and methods: Cloning for in planta assays |
| Recombinant DNA reagent | pICSL13004 | TSL (The Sainsbury Laboratory) SynBio team | | Mas promoter Golden Gate module |
| Recombinant DNA reagent | pICSL50001 | TSL (The Sainsbury Laboratory) SynBio team | | C-terminal HF Golden Gate module |
| Recombinant DNA reagent | pICH77901 | TSL (The Sainsbury Laboratory) SynBio team | | Mas terminator Golden Gate module |
| Recombinant DNA reagent | p41308-PikmN | This paper | | Materials and methods: Cloning for in planta assays |
| Recombinant DNA reagent | p41308-PikmC | This paper | | Materials and methods: Cloning for in planta assays |
| Recombinant DNA reagent | pOPIN-M | Addgene | No. 26044 | *E. coli* expression vector |
| Recombinant DNA reagent | AVR-PikD in pOPIN-S3C | *Maqbool et al., 2015* | | *E. coli* expression construct |
| Commercial assay, kit | Anti-HA Affinity Matrix, from rat IgG1 | Roche | 11815016001 | Materials and methods: Protein–protein interaction studies: co-IP; 20 µL |
| Antibody | HA-probe (F-7) HRP-conjugated; mouse monoclonal IgG2a | Santa Cruz Biotech | sc-7392 | Materials and methods: Protein–protein interaction studies: co-IP; 1:5000 |
| Antibody | Mouse monoclonal ANTI-FLAG M2 | Sigma | F3165 | Materials and methods: Protein–protein interaction studies: co-IP |
| Antibody | A-14 anti-Myc antibody; A-14 anti-Myc antibody | Santa Cruz Biotechnology | Sc-40 | Materials and methods: Protein–protein interaction studies: co-IP; 1:5000 |
| Commercial assay, kit | Pierce ECL Western Blotting Substrate | Thermo Fisher Scientific | 32109 | Materials and methods: Protein–protein interaction studies: co-IP; 1:5000 |
| Commercial assay, kit | SuperSignal West Femto Maximum Sensitivity Substrate | Thermo Fisher Scientific | 34094 | Materials and methods: Protein–protein interaction studies: co-IP; 1:5000 |

*Continued on next page*

*Continued*

| Reagent type (species) or resource | Designation | Source or reference | Identifiers | Additional information |
|---|---|---|---|---|
| Commercial assay, kit | Pierce Reversible Protein Stain Kit | Thermo Fisher Scientific | 24585 | Materials and methods: Protein–protein interaction studies: co-IP; 1:5000 |
| Software, algorithm | CCP4i2 graphical interface | *Potterton et al., 2018* | | Materials and methods: Crystallisation, data collection, and structure solution |
| Software, algorithm | MolProbity | *Chen et al., 2010* | | Materials and methods: Crystallisation, data collection, and structure solution |
| Software, algorithm | CCP4MG | *McNicholas et al., 2011* | | Materials and methods: Crystallisation, data collection, and structure solution |
| Software, algorithm | SWISS-MODEL | *Waterhouse et al., 2018* | | Materials and methods: Crystallisation, data collection, and structure solution |
| Software, algorithm | besthr | *MacLean, 2019* | | Materials and methods: Cell death assay |
| Software, algorithm | NLR-Parser | *Steuernagel et al., 2015* | | |
| Software, algorithm | HMMER 3.2b2 | *Eddy, 1998* | | Materials and methods: Identification and phylogenetic analysis of CC-NLRs from grasses |
| Software, algorithm | MUSCLE v2.8.31 | *Edgar, 2004* | | Materials and methods: Identification and phylogenetic analysis of CC-NLRs from grasses |
| Software, algorithm | QKphylogeny | https://github.com/matthewmoscou/QKphylogeny | | Materials and methods: Identification and phylogenetic analysis of CC-NLRs from grasses |
| Software, algorithm | RAxML v8.2.11 | *Stamatakis, 2014* | | Materials and methods: Identification and phylogenetic analysis of CC-NLRs from grasses |
| Software, algorithm | iTOL v5.5.1 | *Letunic and Bork, 2007* | | Materials and methods: Identification and phylogenetic analysis of CC-NLRs from grasses |
| Software, algorithm | BLAST v2.3.0 | *Altschul et al., 1990* | | Materials and methods: Identification and phylogenetic analysis of Pik-1 and Pik-2 homologues |
| Software, algorithm | MEGA X | *Kumar et al., 2018* | | Materials and methods: Phylogenetic analyses of rice HMA domains and ancestral sequence reconstruction |
| Software, algorithm | FastML | *Ashkenazy et al., 2012* | | Materials and methods: Phylogenetic analyses of rice HMA domains and ancestral sequence reconstruction |
| Software, algorithm | PAML v4.9j | *Yang, 1997* | | Materials and methods: Testing for selection |
| Software, algorithm | *ggplot2* R v3.6.3 package | *Ginestet, 2011* | | Materials and methods: Testing for selection |
| Software, algorithm | SNAP | https://www.hiv.lanl.gov/ | | Materials and methods: Testing for selection |
| Sequence-based reagent | 5′-TGAAGCAGATCCGAGACATAGCCT-3′ | This study | PCR primer | Materials and methods: Identification and cloning of Pik-1 and Pik-2 from *Oryza brachyantha* |
| Sequence-based reagent | 5′-TACCCTGCTCCTGATTGCTGACT-3′ | This study | PCR primer | Materials and methods: Identification and cloning of Pik-1 and Pik-2 from *Oryza brachyantha* |

*Continued on next page*

*Continued*

| Reagent type (species) or resource | Designation | Source or reference | Identifiers | Additional information |
|---|---|---|---|---|
| Sequence-based reagent | 5′-AGGGAGCAATGATGCTTCACGA-3′ | This study | PCR primer | Materials and methods: Identification and cloning of the Pik-1–integrated HMA domains from wild rice relatives |
| Sequence-based reagent | 3′-TTCTCTGGCAACCGTTGTTTTGC-5′ | This study | PCR primer | Materials and methods: Identification and cloning of the Pik-1–integrated HMA domains from wild rice relatives |
| Commercial assay or kit | In-Fusion HD Cloning | Clontech | 639647 | Materials and methods: Cloning for in vitro studies |
| Gene (*O. brachyantha*) | W0654 | Wild Rice Collection 'Oryzabase'; **Kurata and Yamazaki, 2006** | | Materials and methods: Identification and cloning of Pik-1 and Pik-2 from *Oryza brachyantha* |
| Gene (*O. brachyantha*) | W0655 | Wild Rice Collection 'Oryzabase'; **Kurata and Yamazaki, 2006** | | Materials and methods: Identification and cloning of Pik-1 and Pik-2 from *Oryza brachyantha* |
| Gene (*O. brachyantha*) | W0656 | Wild Rice Collection 'Oryzabase'; **Kurata and Yamazaki, 2006** | | Materials and methods: Identification and cloning of Pik-1 and Pik-2 from *Oryza brachyantha* |
| Gene (*O. brachyantha*) | W1057 | Wild Rice Collection 'Oryzabase'; **Kurata and Yamazaki, 2006** | | Materials and methods: Identification and cloning of Pik-1 and Pik-2 from *Oryza brachyantha* |
| Gene (*O. brachyantha*) | W1401 | Wild Rice Collection 'Oryzabase'; **Kurata and Yamazaki, 2006** | | Materials and methods: Identification and cloning of Pik-1 and Pik-2 from *Oryza brachyantha* |
| Gene (*O. brachyantha*) | W1402 | Wild Rice Collection 'Oryzabase'; **Kurata and Yamazaki, 2006** | | Materials and methods: Identification and cloning of Pik-1 and Pik-2 from *Oryza brachyantha* |
| Gene (*O. brachyantha*) | W1403 | Wild Rice Collection 'Oryzabase'; **Kurata and Yamazaki, 2006** | | Materials and methods: Identification and cloning of Pik-1 and Pik-2 from *Oryza brachyantha* |
| Gene (*O. brachyantha*) | W1404 | Wild Rice Collection 'Oryzabase'; **Kurata and Yamazaki, 2006** | | Materials and methods: Identification and cloning of Pik-1 and Pik-2 from *Oryza brachyantha* |
| Gene (*O. brachyantha*) | W1405 | Wild Rice Collection 'Oryzabase'; **Kurata and Yamazaki, 2006** | | Materials and methods: Identification and cloning of Pik-1 and Pik-2 from *Oryza brachyantha* |
| Gene (*O. brachyantha*) | W1407(B) | Wild Rice Collection 'Oryzabase'; **Kurata and Yamazaki, 2006** | | Materials and methods: Identification and cloning of Pik-1 and Pik-2 from *Oryza brachyantha* |
| Gene (*O. brachyantha*) | W1703 | Wild Rice Collection 'Oryzabase'; **Kurata and Yamazaki, 2006** | | Materials and methods: Identification and cloning of Pik-1 and Pik-2 from *Oryza brachyantha* |
| Gene (*O. brachyantha*) | W1705 | Wild Rice Collection 'Oryzabase'; **Kurata and Yamazaki, 2006** | | Materials and methods: Identification and cloning of Pik-1 and Pik-2 from *Oryza brachyantha* |

*Continued on next page*

*Continued*

| Reagent type (species) or resource | Designation | Source or reference | Identifiers | Additional information |
|---|---|---|---|---|
| Gene (O. brachyantha) | W1706 | Wild Rice Collection 'Oryzabase'; *Kurata and Yamazaki, 2006* | | Materials and methods: Identification and cloning of Pik-1 and Pik-2 from *Oryza brachyantha* |
| Gene (O. brachyantha) | W1708 | Wild Rice Collection 'Oryzabase'; *Kurata and Yamazaki, 2006* | | Materials and methods: Identification and cloning of Pik-1 and Pik-2 from *Oryza brachyantha* |
| Gene (O. brachyantha) | W1711 | Wild Rice Collection 'Oryzabase'; *Kurata and Yamazaki, 2006* | | Materials and methods: Identification and cloning of Pik-1 and Pik-2 from *Oryza brachyantha* |
| Gene (O. brachyantha) | W1712 | Wild Rice Collection 'Oryzabase'; *Kurata and Yamazaki, 2006* | | Materials and methods: Identification and cloning of Pik-1 and Pik-2 from *Oryza brachyantha* |
| Gene (O. brachyantha) | W0654 | Wild Rice Collection 'Oryzabase'; *Kurata and Yamazaki, 2006* | | Materials and methods: Identification and cloning of the Pik-1-integrated HMA domains from wild rice relatives |
| Gene (O. australiensis) | W0008 | Wild Rice Collection 'Oryzabase'; *Kurata and Yamazaki, 2006* | | Materials and methods: Identification and cloning of the Pik-1-integrated HMA domains from wild rice relatives |
| Gene (O. australiensis) | W1628 | Wild Rice Collection 'Oryzabase'; *Kurata and Yamazaki, 2006* | | Materials and methods: Identification and cloning of the Pik-1-integrated HMA domains from wild rice relatives |
| Gene (O. barthii) | W1643 | Wild Rice Collection 'Oryzabase'; *Kurata and Yamazaki, 2006* | | Materials and methods: Identification and cloning of the Pik-1-integrated HMA domains from wild rice relatives |
| Gene (O. barthii) | W1605 | Wild Rice Collection 'Oryzabase'; *Kurata and Yamazaki, 2006* | | Materials and methods: Identification and cloning of the Pik-1-integrated HMA domains from wild rice relatives |
| Gene (O. barthii) | W0042 | Wild Rice Collection 'Oryzabase'; *Kurata and Yamazaki, 2006* | | Materials and methods: Identification and cloning of the Pik-1-integrated HMA domains from wild rice relatives |
| Gene (O. barthii) | W0698 | Wild Rice Collection 'Oryzabase'; *Kurata and Yamazaki, 2006* | | Materials and methods: Identification and cloning of the Pik-1-integrated HMA domains from wild rice relatives |
| Gene (O. eichingeri) | W1526 | Wild Rice Collection 'Oryzabase'; *Kurata and Yamazaki, 2006* | | Materials and methods: Identification and cloning of the Pik-1-integrated HMA domains from wild rice relatives |
| Gene (O. glumaepatula) | W1171 | Wild Rice Collection 'Oryzabase'; *Kurata and Yamazaki, 2006* | | Materials and methods: Identification and cloning of the Pik-1-integrated HMA domains from wild rice relatives |
| Gene (O. glumaepatula) | W2203 | Wild Rice Collection 'Oryzabase'; *Kurata and Yamazaki, 2006* | | Materials and methods: Identification and cloning of the Pik-1-integrated HMA domains from wild rice relatives |
| Gene (O. grandiglumis) | W1480(B) | Wild Rice Collection 'Oryzabase'; *Kurata and Yamazaki, 2006* | | Materials and methods: Identification and cloning of the Pik-1-integrated HMA domains from wild rice relatives |

*Continued on next page*

*Continued*

| Reagent type (species) or resource | Designation | Source or reference | Identifiers | Additional information |
|---|---|---|---|---|
| Gene (*O. granulata*) | W0005 | Wild Rice Collection 'Oryzabase'; *Kurata and Yamazaki, 2006* | | Materials and methods: Identification and cloning of the Pik-1-integrated HMA domains from wild rice relatives |
| Gene (*O. granulata*) | W0067(B) | Wild Rice Collection 'Oryzabase'; *Kurata and Yamazaki, 2006* | | Materials and methods: Identification and cloning of the Pik-1-integrated HMA domains from wild rice relatives |
| Gene (*O. latifolia/O. alta*) | W0542 | Wild Rice Collection 'Oryzabase'; *Kurata and Yamazaki, 2006* | | Materials and methods: Identification and cloning of the Pik-1-integrated HMA domains from wild rice relatives |
| Gene (*O. latifolia/O. alta*) | W1539 | Wild Rice Collection 'Oryzabase'; *Kurata and Yamazaki, 2006* | | Materials and methods: Identification and cloning of the Pik-1-integrated HMA domains from wild rice relatives |
| Gene (*O. longiglumis*) | W1228 | Wild Rice Collection 'Oryzabase'; *Kurata and Yamazaki, 2006* | | Materials and methods: Identification and cloning of the Pik-1-integrated HMA domains from wild rice relatives |
| Gene (*O. longistaminata*) | W1504 | Wild Rice Collection 'Oryzabase'; *Kurata and Yamazaki, 2006* | | Materials and methods: Identification and cloning of the Pik-1-integrated HMA domains from wild rice relatives |
| Gene (*O. longistaminata*) | W1540 | Wild Rice Collection 'Oryzabase'; *Kurata and Yamazaki, 2006* | | Materials and methods: Identification and cloning of the Pik-1-integrated HMA domains from wild rice relatives |
| Gene (*O. longistaminata*) | W0643 | Wild Rice Collection 'Oryzabase'; *Kurata and Yamazaki, 2006* | | Materials and methods: Identification and cloning of the Pik-1-integrated HMA domains from wild rice relatives |
| Gene (*O. meridionalis*) | W2081 | Wild Rice Collection 'Oryzabase'; *Kurata and Yamazaki, 2006* | | Materials and methods: Identification and cloning of the Pik-1-integrated HMA domains from wild rice relatives |
| Gene (*O. meridionalis*) | W2112 | Wild Rice Collection 'Oryzabase'; *Kurata and Yamazaki, 2006* | | Materials and methods: Identification and cloning of the Pik-1-integrated HMA domains from wild rice relatives |
| Gene (*O. meyeriana*) | W1354 | Wild Rice Collection 'Oryzabase'; *Kurata and Yamazaki, 2006* | | Materials and methods: Identification and cloning of the Pik-1-integrated HMA domains from wild rice relatives |
| Gene (*O. minuta*) | W1328 | Wild Rice Collection 'Oryzabase'; *Kurata and Yamazaki, 2006* | | Materials and methods: Identification and cloning of the Pik-1-integrated HMA domains from wild rice relatives |
| Gene (*O. officinalis*) | W0614 | Wild Rice Collection 'Oryzabase'; *Kurata and Yamazaki, 2006* | | Materials and methods: Identification and cloning of the Pik-1-integrated HMA domains from wild rice relatives |
| Gene (*O. officinalis*) | W1200 | Wild Rice Collection 'Oryzabase'; *Kurata and Yamazaki, 2006* | | Materials and methods: Identification and cloning of the Pik-1-integrated HMA domains from wild rice relatives |
| Gene (*O. punctata*) | W1408 | Wild Rice Collection 'Oryzabase'; *Kurata and Yamazaki, 2006* | | Materials and methods: Identification and cloning of the Pik-1-integrated HMA domains from wild rice relatives |

*Continued on next page*

*Continued*

| Reagent type (species) or resource | Designation | Source or reference | Identifiers | Additional information |
|---|---|---|---|---|
| Gene (*O. punctata*) | W1514 | Wild Rice Collection 'Oryzabase'; ***Kurata and Yamazaki, 2006*** | | Materials and methods: Identification and cloning of the Pik-1-integrated HMA domains from wild rice relatives |
| Gene (*O. rhizomatis*) | W1808 | Wild Rice Collection 'Oryzabase'; ***Kurata and Yamazaki, 2006*** | | Materials and methods: Identification and cloning of the Pik-1-integrated HMA domains from wild rice relatives |
| Gene (*O. ridleyi*) | W0001 | Wild Rice Collection 'Oryzabase'; ***Kurata and Yamazaki, 2006*** | | Materials and methods: Identification and cloning of the Pik-1-integrated HMA domains from wild rice relatives |
| Gene (*O. ridleyi*) | W2035 | Wild Rice Collection 'Oryzabase'; ***Kurata and Yamazaki, 2006*** | | Materials and methods: Identification and cloning of the Pik-1-integrated HMA domains from wild rice relatives |
| Gene (*O. rufipogon*) | W2003 | Wild Rice Collection 'Oryzabase'; ***Kurata and Yamazaki, 2006*** | | Materials and methods: Identification and cloning of the Pik-1-integrated HMA domains from wild rice relatives |
| Gene (*O. rufipogon*) | W1715 | Wild Rice Collection 'Oryzabase'; ***Kurata and Yamazaki, 2006*** | | Materials and methods: Identification and cloning of the Pik-1-integrated HMA domains from wild rice relatives |
| Gene (*O. rufipogon/ O. meridionalis*) | W2117 | Wild Rice Collection 'Oryzabase'; ***Kurata and Yamazaki, 2006*** | | Materials and methods: Identification and cloning of the Pik-1-integrated HMA domains from wild rice relatives |
| Gene (*O. brachyantha*) | LOC102699268 | GenBank | | Materials and methods: Phylogenetic analyses of rice HMA domains and ancestral sequence reconstruction |
| Gene (*O. barthii*) | OBART11G23150 | GenBank | | Materials and methods: Phylogenetic analyses of rice HMA domains and ancestral sequence reconstruction |
| Gene (*O. longistaminata*) | KN541092.1 | GenBank | | Materials and methods: Phylogenetic analyses of rice HMA domains and ancestral sequence reconstruction |
| Gene (*O. punctata*) | OPUNC11G19550 | GenBank | | Materials and methods: Phylogenetic analyses of rice HMA domains and ancestral sequence reconstruction |
| Gene (*O. sativa*) | HM035360.1 | GenBank | | Materials and methods: Phylogenetic analyses of rice HMA domains and ancestral sequence reconstruction |
| Gene (*O. sativa*) | HM048900_1 | GenBank | | Materials and methods: Phylogenetic analyses of rice HMA domains and ancestral sequence reconstruction |
| Gene (*O. sativa*) | HQ662330_1 | GenBank | | Materials and methods: Phylogenetic analyses of rice HMA domains and ancestral sequence reconstruction |
| Gene (*O. sativa*) | HQ662329_1 | GenBank | | Materials and methods: Phylogenetic analyses of rice HMA domains and ancestral sequence reconstruction |
| Gene (*O. sativa*) | AB462324.1 | GenBank | | Materials and methods: Phylogenetic analyses of rice HMA domains and ancestral sequence reconstruction |
| Gene (*O. brachyantha*) | LOC102708959 | GenBank | | Materials and methods: Phylogenetic analyses of rice HMA domains and ancestral sequence reconstruction |

Continued

| Reagent type (species) or resource | Designation | Source or reference | Identifiers | Additional information |
|---|---|---|---|---|
| Gene (*O. brachyantha*) | LOC102709146 | GenBank | | Materials and methods: Phylogenetic analyses of rice HMA domains and ancestral sequence reconstruction |
| Gene (*O. brachyantha*) | LOC102714171 | GenBank | | Materials and methods: Phylogenetic analyses of rice HMA domains and ancestral sequence reconstruction |
| Gene (*O. brachyantha*) | LOC102716957 | GenBank | | Materials and methods: Phylogenetic analyses of rice HMA domains and ancestral sequence reconstruction |
| Gene (*O. brachyantha*) | LOC102717220 | GenBank | | Materials and methods: Phylogenetic analyses of rice HMA domains and ancestral sequence reconstruction |
| Gene (*O. sativa*) | LOC_Os04g39360 | GenBank | | Materials and methods: Phylogenetic analyses of rice HMA domains and ancestral sequence reconstruction |
| Gene (*O. sativa*) | LOC_Os04g39370 | GenBank | | Materials and methods: Phylogenetic analyses of rice HMA domains and ancestral sequence reconstruction |
| Gene (*O. sativa*) | Os04g0469000_01 | GenBank | | Materials and methods: Phylogenetic analyses of rice HMA domains and ancestral sequence reconstruction |
| Gene (*O. sativa*) | Os02g0585200 | GenBank | | Materials and methods: Phylogenetic analyses of rice HMA domains and ancestral sequence reconstruction |
| Gene (*O. sativa*) | Os02g0584800_01 | GenBank | | Materials and methods: Phylogenetic analyses of rice HMA domains and ancestral sequence reconstruction |
| Gene (*O. sativa*) | Os02g0584700_01 | GenBank | | Materials and methods: Phylogenetic analyses of rice HMA domains and ancestral sequence reconstruction |
| Gene (*O. sativa*) | Os04g0469300_01 | GenBank | | Materials and methods: Phylogenetic analyses of rice HMA domains and ancestral sequence reconstruction |
| Gene (*O. sativa*) | Os02g0585100 | GenBank | | Materials and methods: Phylogenetic analyses of rice HMA domains and ancestral sequence reconstruction |
| Gene (*O. sativa*) | Os02g0584600 | GenBank | | Materials and methods: Phylogenetic analyses of rice HMA domains and ancestral sequence reconstruction |
| Gene (*O. sativa*) | OSJNBa0060P14.7_01 | GenBank | | Materials and methods: Phylogenetic analyses of rice HMA domains and ancestral sequence reconstruction |
| Gene (*O. sativa*) | Os04g0464100_01 | GenBank | | Materials and methods: Phylogenetic analyses of rice HMA domains and ancestral sequence reconstruction |
| Gene (*O. sativa*) | Os02g0582600 | GenBank | | Materials and methods: Phylogenetic analyses of rice HMA domains and ancestral sequence reconstruction |

### Identification and phylogenetic analysis of CC-NLRs from grasses

NLR-parser (*Steuernagel et al., 2015*) was used to identify the NLR sequences from the predicted protein databases of eight representative grass species, *Brachypodium distachyon*, *O. brachyantha*, *Oryza sativa*, *S. bicolor*, *Triticum aestivum*, *Zea mays* (downloaded from Ensembl Plants collection), and *Hordeum vulgare* and *S. italica* (downloaded from Phytozome v12.1 collection), listed in *Supplementary file 1A*. NLR sequences that were longer than 750 amino acid were screened for features of the NB-ARC and LRR domains, defined by the PF00931, PF00560, PF07725, PF13306, and PF13855 pfam models, using HMMER 3.2b2 (*Eddy, 1998*); signatures of the coiled-coil domain

were identified using 'motif16' and 'motif17' defined in NLR-parser. Protein sequences of NLRs that contained at least two of the above features were aligned using MUSCLE v2.8.31 (*Edgar, 2004*). The proteins comprising fewer than 60 amino acids N- and C-terminally of the NB-ARC domain, relative to the NB-ARC domain of Pikp-2 (*Maqbool et al., 2015*), were removed, as were sequences with less than 50% coverage across the alignment. The dataset was further filtered so that for each gene there was only one representative protein isoform—with the exception of sequences from *B. distachyon* and *S. bicolor* that did not carry gene identifiers. Filtering resulted in a final list of 3062 CC-NLRs (*Supplementary file 2* ) that were amended with 35 known and functionally characterised NLR-type resistance proteins from grasses, added for reference (*Supplementary file 1B*).

The amino acid sequences corresponding to the NB-ARC domain of the identified NLRs were aligned using MUSCLE v2.8.31 (*Edgar, 2004*). The alignment positions with more than 30% data missing were removed from the alignment using QKphylogeny (*Moscou, 2019*; https://github.com/matthewmoscou/QKphylogeny). This revealed a final alignment of 241 amino acids, which was used for a phylogenetic analysis. A ML phylogenetic tree was calculated using RAxML v8.2.11 (*Stamatakis, 2014*) with bootstrap values (*Felsenstein, 1985*) based on 1000 iterations and best-scoring JTT likelihood model (*Jones et al., 1992*) selected by automatic protein model assignment using the ML criterion. Best ML tree was mid-point rooted and visualised using Interactive Tree of Life (iTOL) tool v5.5.1 (*Letunic and Bork, 2007*). The relationships of 28 and 38 proteins that grouped with rice Pikp-1 and Pikp-2, respectively, were further validated as follows. Genetic loci and gene coordinates for each of those NLRs were inspected and, if required, manually reannotated; identifiers of manually reannotated genes were amended with '.n' suffix. For each gene, one splice version was selected and aligned using MUSCLE v2.8.31 (*Edgar, 2004*). The ML phylogenetic trees of Pik-1- and Pik-2-related NLRs were calculated based on positions within the NB-ARC domain, for which more than 70% of data were present—957 and 1218 nucleotides for Pik-1 and Pik-2, respectively. The trees were generated using RAxML v8.2.11 (*Stamatakis, 2014*) with bootstrap values (*Felsenstein, 1985*) based on 1000 iterations and GTRGAMMA substitution model (*Tavaré, 1986*). Best ML trees were manually rooted based on the relationships observed in the above analyses and visualised using the iTOL tool v5.5.1 (*Letunic and Bork, 2007*).

## Identification and phylogenetic analysis of Pik-1 and Pik-2 homologues

Coding sequences of representative Pik-1 and Pik-2 genes were used to identify Pik homologues from cDNA databases of *Oryza barthii*, *Oryza longistaminata*, *Oryza punctata*, *Oryza glumeapatula*, *Oryza glaberrima*, *Oryza rufipogon*, *Oryza nivara*, *L. perrieri*, *Zizania latifolia*, and *Dactylis glomerata*, listed in *Supplementary file 1D*, using BLAST v2.3.0 (*Altschul et al., 1990*). For each sequence with BLASTN E-value cutoff <0.01, genetic loci and gene coordinates were inspected and, if necessary, manually reannotated; identifiers of manually reannotated genes were amended with '.n' suffix. Because the *Pik-1* and *Pik-2* genes are known to be genetically linked, each *Pik* locus was further examined for signatures of unpredicted *Pik* gene candidates. Next, coding sequences of the Pik-1 and Pik-2 candidate homologues were aligned using MUSCLE v2.8.31 (*Edgar, 2004*). Poorly aligned sequences were manually removed from the alignment and excluded from further analysis. The phylogenetic trees were calculated based on positions within the NB-ARC domain, for which more than 70% of data was present—927 and 1239 nucleotides of 46 Pik-1 and 54 Pik-2 candidates, respectively. ML phylogenetic trees were calculated using RAxML v8.2.11 (*Stamatakis, 2014*) with bootstrap values based on 1000 iterations (*Felsenstein, 1985*) and GTRGAMMA substitution model (*Tavaré, 1986*). Best ML trees were manually rooted according to previously observed relationship and visualised using the iTOL tool v5.5.1 (*Letunic and Bork, 2007*).

## Phylogenetic analyses of rice HMA domains and ancestral sequence reconstruction

Selected non-integrated HMA sequences from *O. sativa* and *O. brachyantha* were obtained by BLASTP search (*Altschul et al., 1990*) using Pikp-1 HMA (Pikp-HMA) as a query. Amino acid and nucleotide alignments were generated using MUSCLE (*Edgar, 2004*). NJ clustering method (*Saitou and Nei, 1987*) was used for constructing protein-based or codon-based trees based on JTT (*Jones et al., 1992*) or Maximum Composite Likelihood substitution models, respectively, using 1000 bootstrap tests (*Felsenstein, 1985*), as implemented in MEGA X (*Kumar et al., 2018*). ML

trees were calculated using JTT (*Jones et al., 1992*) or GTR (*Tavaré, 1986*) substitution models as implemented in MEGA X software (*Kumar et al., 2018*).

Three independent protein sequence alignments, generated with MUSCLE (*Edgar, 2004*), were used for ASR (*Supplementary file 1M*). Joint and marginal ASRs were performed with FastML software (*Ashkenazy et al., 2012*) using JTT substitution model (*Jones et al., 1992*), gamma distribution, and 90% probability cutoff to prefer ancestral indel over a character. The reconstruction was performed based on NJ trees (*Saitou and Nei, 1987*) built with 100 iteration bootstrap method (*Felsenstein, 1985*). Sequences after marginal reconstruction including indels were used for further analyses.

## Testing for selection

The rates of synonymous ($d_S$) and nonsynonymous ($d_N$) nucleotide substitutions per site in pairwise comparisons of protein-coding DNA sequences were estimated using the *Yang and Nielsen, 2000* method under realistic evolutionary models, as implemented in the YN00 program in the PAML v4.9j package (*Yang, 1997*). The coding sequence alignments used for the analysis were generated using MUSCLE v2.8.31 (*Edgar, 2004*); unless stated otherwise, only positions that showed over 70% coverage across the alignment were used for the analyses.

For selection across the sites of the HMA domain, site models were implemented using the CODEML program in the PAML v4.9j software package (*Yang, 1997*). The three null models, M0 (one-ratio), M1 (nearly neutral), M7 (beta), and three alternative models, M3 (selection), M2 (discrete), M8 (beta and ω), were tested as recommended by *Yang et al., 2000*, and their likelihoods were calculated with the LRT. The difference in log likelihood ratio between a null model and an alternative model was multiplied by 2 and compared with the chi-squared ($\chi^2$) distribution; the degrees of freedom were calculated from the difference in the numbers of parameters estimated from the model pairs. The naïve empirical Bayes (NEB) (*Yang, 2000*; *Yang and Nielsen, 1998*) or the BEB (*Yang et al., 2005*) were used to infer the posterior probabilities for site classes and identify amino acids under positive selection. Raw data were extracted and visualised using the *ggplot2* R v3.6.3 package (*Ginestet, 2011*). ML phylogenetic tree used for the analysis was built with bootstrap values (*Felsenstein, 1985*) from 1000 iterations using MEGA X software (*Kumar et al., 2018*), based on coding sequence alignment, generated with MUSCLE v2.8.31 (*Edgar, 2004*).

The pairwise rates of synonymous and nonsynonymous substitutions across Pik-1 allelic variants of rice were calculated using the *Nei and Gojobori, 1986* method, as implemented using the SNAP tool (https://www.hiv.lanl.gov/).

## Identification and cloning of *Pik-1* and *Pik-2* from *O. brachyantha*

Genomic DNA materials of 16 *O. brachyantha* accessions were ordered from Wild Rice Collection 'Oryzabase' (*Supplementary file 1C*; *Kurata and Yamazaki, 2006*). The accessions were first screened for deletion within the *Pik-2* gene, present in a reference genome of *O. brachyantha* (*Chen et al., 2013*). Selected accessions were used to amplify full-length *Pik-1* and *Pik-2* genes using 5′-TGAAGCAGATCCGAGACATAGCCT-3′ and 5′-TACCCTGCTCCTGATTGCTGACT-3′ primers designed based on the *O. brachyantha* genome sequence (*Chen et al., 2013*). The PCRs were run on agarose gels to check amplification and product size against positive controls. Fragments of the expected size were further gel purified, cloned into Zero Blunt TOPO plasmid (Thermo Fisher Scientific), and sequenced.

## Identification and cloning of the Pik-1-integrated HMA domains from wild rice relatives

Genomic DNA materials of 1–3 accessions of 18 wild rice species—*Oryza australiensis*, *O. barthii*, *O. brachyantha*, *Oryza eichingeri*, *Oryza glumaepatula*, *Oryza grandiglumis*, *Oryza granulata*, *Oryza latifolia*, *Oryza longiglumis*, *O. longistaminata*, *Oryza meridionalis*, *Oryza meyeriana*, *Oryza minuta*, *Oryza officinalis*, *O. punctata*, *Oryza rhizomatis*, *Oryza ridleyi*, *O. rufipogon*—were ordered from Wild Rice Collection 'Oryzabase' (*Kurata and Yamazaki, 2006*) and used for amplification of the Pik-1-integrated HMA (*Supplementary file 1H*). The 5′-AGGGAGCAATGATGCTTCACGA-3′ and 3′-TTCTCTGGCAACCGTTGTTTTGC-5′ primers were designed using the alignment of the *OsPikp-1* and *OBRAC11G13570.1* sequences and used in PCR. The amplicons were run on agarose gels to

check amplification and product sizes against positive controls. Fragments of 450–720 bp in size were gel-purified, cloned into Zero Blunt TOPO plasmid (Thermo Fisher Scientific), and sequenced. Genotyping was performed twice, and only sequences that did not show ambiguity between sequencing runs were selected for further analyses.

## Cloning for in planta assays

The rice Pikp-1, previously cloned by *Maqbool et al., 2015*, was amplified from pCambia1300:AscI plasmid and domesticated to remove internal *BsaI* and *BpiI* restriction enzyme recognition sites using site-directed mutagenesis by inverse PCR. The amplicons were purified and assembled using the Golden Gate method (*Weber et al., 2011*) in the level 0 pICH41308 (Addgene no. 47998) destination vector for subsequent Golden Gate cloning. The N-terminally tagged HA:Pikp-1 expression construct was generated by Golden Gate assembly with pICSL12008 (35S + Ω promoter, TSL SynBio), pICSL30007 (N-terminal 6×HA, TSL SynBio), and pICH41414 (35S terminator, Addgene no. 50337) modules, into the binary vector pICH47732 (Addgene no. 48001). Using the same set of Golden Gate modules, Pikp-1$_{E230R}$ mutant was subcloned into the same binary vector, generating the N-terminally tagged HA:Pikp-1$_{E230R}$ expression construct.

The ancHMA variants—corresponding to 186–260 residues of the full-length Pikp-1—were synthesised as level 0 modules for Golden Gate cloning by GENEWIZ (South Plainfield, NJ, USA). Cloning of subsequent Pikp-1:ancHMA fusions was done using two custom-made Golden Gate level 0 acceptor plasmids, p41308-PikpN and p41308-PikpC, that allowed HMA insertion in a single Golden Gate level 0 reaction, generating full-length Pikp-1 constructs with or without a stop codon, respectively. The ancHMA mutants—ancHMA$_{AMEGNND}$, ancHMA$_{LY}$, ancHMA$_{PI}$, ancHMA$_{LVKIE}$, and the single mutants within the LVKIE region of the ancHMA—were synthesised by GENEWIZ and subcloned into p41308-PikpN and p41308-PikpC plasmids for cloning. Two of the ancHMA mutants, ancHMA$_{IVQVE}$ and ancHMA$_{LVKIV}$, were generated using site-directed mutagenesis by inverse PCR and cloned into the same acceptor plasmids. Using the p41308-PikpN modules, HA:Pikp-1:ancHMA expression constructs were generated by Golden Gate assembly with pICSL12008 (35S + Ω promoter, TSL SynBio), pICSL30007 (N-terminal 6×HA, TSL SynBio), and pICH41414 (35S terminator, Addgene no. 50337) into the binary vector pICH47732 (Addgene no. 48001). To generate C-terminally tagged expression constructs, the p41308-PikpC modules were assembled with pICSL13004 (Mas promoter, TSL SynBio), pICSL50001 (C-terminal HF, TSL SynBio), and pICH77901 (Mas terminator, TSL SynBio) by Golden Gate method into the same binary vector.

To generate Pikm-1:ancHMA fusions, ancHMA N2-I, ancHMA$_{EMVKE}$, ancHMA$_{FFE}$, ancHMA$_{STSN}$, ancHMA$_{VH}$, and ancHMA$_{IVDPM}$ were synthesised by GENEWIZ as Golden Gate modules. The ancHMA$_{EMANK}$ mutant was generated by amplification and fusion of the N-terminus of ancHMA$_{EMVKE}$ construct and the C-terminus of N2-I ancHMA variant. All ancHMA constructs corresponded to 187–264 residues of the full-length Pikm-1 protein and were subsequently assembled with custom-made p41308-PikmN (TSL SynBio) or p41308-PikmC (TSL SynBio) level 0 acceptors to generate Pikm-1:ancHMA fusions with or without a stop codon, respectively. Obtained modules were then used to generate Pikm-1:ancHMA expression constructs, featuring either N-terminal HA of C-terminal HF tags, by Golden Gate assembly using the same set of modules as previously used for Pikp-1 and pICH47732 binary vector.

## Cloning for in vitro studies

The ancHMA mutants were amplified from Golden Gate level 0 modules by PCR and cloned into pOPIN-M vector featuring N-terminal 6xHis and MBP tags with a 3C protease cleavage site using In-Fusion cloning (*Berrow et al., 2007*). The AVR-PikD used for crystallography was cloned into pOPIN-S3C featuring N-terminal 6xHis and SUMO tags with a 3C protease cleavage site using In-Fusion reaction. AVR-PikD used for SPR studies was cloned previously (*Maqbool et al., 2015*).

## Protein–protein interaction studies: co-IP

The co-IP protocol was described previously (*Win et al., 2011*). Transient gene expression in planta was conducted by delivering T-DNA constructs within *Agrobacterium tumefaciens* strain GV3101::pMP90 into *N. benthamiana* leaves, and the leave tissue was collected 3 days after infiltration. Co-IP was performed using affinity chromatography with anti-HA Affinity Matrix (Roche). After co-IP and

washing, the beads were resuspended in 30 μL of loading dye and eluted by incubating at 70°C for 10 min. Proteins were separated by SDS-PAGE and transferred onto a polyvinylidene difluoride (PVDF) membrane using a Trans-Blot turbo transfer system (Bio-Rad). The membrane was blocked with 5% non-fat dried milk powder in Tris-buffered saline and 1% Tween 20 and probed with appropriate antisera. HA-probe (F-7) horseradish peroxidase (HRP)-conjugated (Santa Cruz Biotech) was used for a single-step detection of HA tag. FLAG detection was carried using monoclonal ANTI-FLAG M2 (Sigma) and anti-mouse HRP-conjugated antibodies in a two-step FLAG detection. A two-step detection of Myc was performed using anti-Myc (A-14, Santa Cruz Biotechnology) and anti-rabbit HRP-conjugated antibodies. Pierce ECL Western Blotting Substrate (Thermo Fisher Scientific) or SuperSignal West Femto Maximum Sensitivity Substrate (Thermo Fisher Scientific) were used for detection. Membranes were imaged using ImageQuant LAS 4000 luminescent imager (GE Healthcare Life Sciences). Equal loading was checked by staining PVDF membranes with Pierce Reversible Protein Stain Kit (Thermo Fisher Scientific), Ponceau S, or Coomassie Brilliant Blue staining solutions.

## Protein–protein interaction studies: SPR

SPR experiments to investigate the effects of the IAQVV/LVKIE and MKANK/EMVKE regions were performed in the SPR buffer 1 (50 mM HEPES, pH 7.5; 300 mM NaCl; and 0.1% Tween 20) and SPR buffer 2 (50 mM HEPES, pH 7.5; 820 mM NaCl; and 0.1% Tween 20), respectively, at 25°C using Biacore T200 (GE Healthcare). The 6xHis-tagged AVR-PikD (ligand) was immobilised on the Series S Sensor Chip NTA (GE Healthcare) and the HMA constructs (analytes) flowed over the effector at a flow rate of 30 μL/min. For each cycle, the chip was washed with the appropriate SPR buffer and activated with 30 μL of 0.5 mM NiCl prior to immobilisation of AVR-PikD. The HMA proteins were injected over both reference and sample cells at a range of concentrations for 120 s, and buffer only flowed for 120 s to record the dissociation. Between each cycle, the sensor chip was regenerated with 30 μL of 0.35 M EDTA. To correct for bulk refractive index changes or machine errors, for each measurement the response was subtracted by the response in the reference cell and the response in buffer-only run (*Myszka, 1999*). The resulting sensorgrams were analysed using the Biacore Insight Evaluation Software (GE Healthcare).

The theoretical maximum responses ($R_{max}$) normalised for the amount of ligand immobilised on the chip were calculated, and the level of binding was expressed as a percentage of $R_{max}$ (%$R_{max}$). Each experiment was repeated a minimum of three times. The data were visualised using *ggplot2* R package (*Ginestet, 2011*).

## Heterologous protein production and purification

Heterologous production and purification of ancHMA were performed as previously described (*Varden et al., 2019*). AVR-PikD and ancHMA proteins used for purification were expressed in pOPIN-S3C and pOPIN-M plasmids, respectively. AVR-PikD effector with non-cleavable C-terminal 6xHis tag, used in SPR, was produced and purified as previously described (*Maqbool et al., 2015*). Protein intact masses were measured by static infusion of samples desalted by acetone precipitation and dissolved in 0.2% formic acid in 30% acetonitrile on Orbitrap Fusion (Thermo Scientific, UK). Data were acquired in a positive mode at 240,000 resolution and 1.6–2 kV spray voltage. The selected spectra were deisotoped and deconvoluted with Xtract software integrated in the Xcalibur package (Thermo Scientific).

## Crystallisation, data collection, and structure solution

Crystallisation screens were performed at 18°C using the sitting-drop vapour diffusion technique. Drops composed of 0.3 μL of protein solution and 0.3 μL of reservoir solution were set up in MRC 96-well crystallisation plates (Molecular Dimensions), which were dispensed using an Oryx Nano or an Oryx8 robot (Douglas Instruments). Crystal growth was monitored using a Minstrel Desktop Crystal Imaging System (Rikagu). We attempted crystallisation of the ancHMA, ancHMA$_{LVKIE}$, and ancHMA$_{EMVKE}$ domains in complexes with AVR-PikD, but only obtained diffracting crystals for ancHMA$_{LVKIE}$–AVR-PikD. These crystals grew after 24–48 hr in 14% (w/v) PEG 3350 and 0.2 M tri-sodium citrate and were harvested into a cryoprotectant comprising the precipitant augmented with 25% (v/v) ethylene glycol before flash-cooling in liquid nitrogen using LithoLoops (Molecular Dimensions). X-ray datasets were collected at the Diamond Light Source using beamline I03 (Didcot, UK)

using a Pilatus3 6M hybrid photon counting detector (Dectris), with crystals maintained at 100 K by a Cryojet cryocooler (Oxford Instruments).

X-ray datasets were integrated and scaled using the DIALS xia2 pipeline (*Winter, 2010*) and merged with AIMLESS (*Evans and Murshudov, 2013*) implemented in the CCP4i2 graphical user interface (*Potterton et al., 2018*), with the best dataset being processed to 1.32 Å resolution in space group $P4_12_12$ with cell parameters $a = b = 119.5$ Å, $c = 36.0$ Å. Since the latter was isomorphous to the HMA–AVR-PikD complex previously solved (PDB accession code 5A6W, *Maqbool et al., 2015*), a high-quality preliminary model could straightforwardly be obtained by direct refinement of the latter against the new dataset using REFMAC5 (*Murshudov et al., 2011*). The asymmetric unit of this preliminary model comprised one copy of AVR-PikD and two copies of ancHMA$_{LVKIE}$. The sequences of the latter chains were subsequently corrected by manually editing the model in COOT (*Emsley et al., 2010*). This model was finalised by iterative rounds of manual rebuilding in COOT (*Emsley et al., 2010*) and restrained refinement with anisotropic thermal parameters in REFMAC5 (*Murshudov et al., 2011*). The resultant structure was assessed with the tools provided in COOT and MolProbity (*Chen et al., 2010*) and visualised using CCP4MG software (*McNicholas et al., 2011*).

## Homology modelling

Homology modelling of the ancHMA structure in complex with AVR-PikD was built using SWISS-MODEL (*Waterhouse et al., 2018*) using coordinates of Pikm-HMA–AVR-PikD structure (PDB accession 6fu9) as a template.

## Cell death assay

Expression constructs and conditions used for cell death/HR assay are listed in *Supplementary file 1N*. Transient expression in *N. benthamiana* leaves was conducted as previously described (*Bos et al., 2006*). Briefly, GV3101::pM90 *A. tumefaciens* strains carrying the appropriate expression vectors were mixed and resuspended in infiltration buffer (10 mM 2-[N-morpholine]-ethanesulfonic acid [MES]; 10 mM MgCl$_2$; and 150 µM acetosyringone, pH 5.6) to a desired density. Upper leaves of 4–5-week-old *N. benthamiana* plants were used for infiltration. The HR cell death was scored 5 days after agroinfiltration using a previously published scale (*Segretin et al., 2014*) modified to range from 0 (no visible necrosis) to 7 (confluent necrosis).

## Acknowledgements

We are thankful to several colleagues, in particular members of the Kamoun and Banfield labs, for discussions, ideas, and support. We thank Dan MacLean and other members of the TSL Bioinformatics team and Mark Youles of TSL SynBio for invaluable technical support. We also thank the Diamond Light Source, UK (beamline I03 under proposal MX18565), for access to X-ray data collection facilities and Andrew Davis and Phil Robinson of JIC Bioimaging facilities for photography. This work was supported by the Gatsby Charitable Foundation, Biotechnology and Biological Sciences Research Council (BBSRC, UK), the European Research Council (ERC grant: proposal 743165) (SK, MJB), and BBSRC Doctoral Training Partnership at Norwich Research Park (grant: BB/M011216/1, project reference: 1771322) (AB). MJB receives funding from BBSRC (BB/P012574, BB/M02198X) and the John Innes Foundation.

## Additional information

### Competing interests

Sophien Kamoun: receives funding from industry on NLR biology. The other authors declare that no competing interests exist.

### Funding

| Funder | Grant reference number | Author |
| --- | --- | --- |
| Gatsby Charitable Foundation | | Sophien Kamoun |

| Biotechnology and Biological Sciences Research Council | | Sophien Kamoun |
|---|---|---|
| Biotechnology and Biological Sciences Research Council | BB/P012574 | Mark J Banfield |
| Biotechnology and Biological Sciences Research Council | BB/M02198X | Mark J Banfield |
| European Research Council | 743165 | Mark J Banfield Sophien Kamoun |
| Biotechnology and Biological Sciences Research Council | BB/M011216/1; 1771322 | Aleksandra Białas |
| John Innes Foundation | | Mark J Banfield |

The funders had no role in study design, data collection and interpretation, or the decision to submit the work for publication.

## Author contributions

Aleksandra Białas, Conceptualization, Resources, Data curation, Formal analysis, Supervision, Funding acquisition, Investigation, Visualization, Methodology, Writing - original draft; Thorsten Langner, Conceptualization, Supervision, Methodology, Writing - original draft; Adeline Harant, Investigation; Mauricio P Contreras, Investigation, Writing - original draft; Clare EM Stevenson, David M Lawson, Jan Sklenar, Data curation, Formal analysis, Methodology, Writing - original draft; Ronny Kellner, Methodology; Matthew J Moscou, Resources, Methodology, Writing - original draft; Ryohei Terauchi, Resources, Funding acquisition, Methodology; Mark J Banfield, Resources, Supervision, Funding acquisition, Writing - original draft; Sophien Kamoun, Conceptualization, Resources, Supervision, Funding acquisition, Visualization, Writing - original draft, Project administration

## Author ORCIDs

Aleksandra Białas (iD) https://orcid.org/0000-0002-1135-2189
Thorsten Langner (iD) https://orcid.org/0000-0002-3401-8888
Mauricio P Contreras (iD) https://orcid.org/0000-0001-6002-0730
David M Lawson (iD) http://orcid.org/0000-0002-7637-4303
Ronny Kellner (iD) http://orcid.org/0000-0002-4618-0110
Matthew J Moscou (iD) https://orcid.org/0000-0003-2098-6818
Mark J Banfield (iD) http://orcid.org/0000-0001-8921-3835
Sophien Kamoun (iD) https://orcid.org/0000-0002-0290-0315

## Decision letter and Author response

Decision letter https://doi.org/10.7554/eLife.66961.sa1
Author response https://doi.org/10.7554/eLife.66961.sa2

# Additional files

## Supplementary files

- Supplementary file 1. Supplementary tables A–N.
- Supplementary file 2. Full list of all Poaceae NLRs and filtering details.
- Supplementary file 3. Site selection test for K-type HMAs.
- Supplementary file 4. ancHMA prediction probabilities.
- Transparent reporting form

## Data availability

Diffraction data have been deposited in PDB under the accession code 7BNT. Sequencing data have been deposited in GenBank under accession codes: MW553204–MW553215 and MW568030–MW568049. All relevant data generated or analysed during this study are included in the manuscript and supporting files. Source data files have been provided for Figure 1—source data 1, Figure 2—

source data 1, Figure 5—source data 1, Figure 6—source data 1, Figure 6—figure supplement 1— source data, Figure 6—Figure supplement 3—source data 1, Figure 8—source data 1.

The following previously published datasets were used:

| Author(s) | Year | Dataset title | Dataset URL | Database and Identifier |
|---|---|---|---|---|

International Brachypodium Initiative2018Genome sequencing and analysis of the model grass Brachypodium distachyonhttps://plants.ensembl.org/Brachypodium_distachyon/Info/Annotation/Ensembl Plants, GCA_00000550 5.4Mascher M, Gundlach H, Himmelbach A, Beier S, Twardziok SO, Wicker T, RadchukV, Dockter C, Hedley PE, Russell J, Bayer M, Ramsay L, Liu H, Haberer G, ZhangXQ, Zhang Q, Barrero RA, Li L, Taudien S, Groth M, Felder M, Hastie A, ŠimkováH, Staňková H, Vrána J, Chan S, Muñoz-Amatriaín M, Ounit R, Wanamaker S, BolserD, Colmsee C, Schmutzer T, Aliyeva-Schnorr L, Grasso S, Tanskanen J, Chailyan A, Sampath D, Heavens D, Clissold L, Cao S, Chapman B, Dai F, Han Y, Li H, Li X, Lin C, McCooke JK, Tan C, Wang P, Wang S, Yin S, Zhou G, Poland JA, Bellgard MI, Borisjuk L, Houben A, Doležel J, Ayling S, Lonardi S, Kersey P, Langridge P, Muehlbauer GJ, Clark MD, Caccamo M, Schulman AH, Mayer KFX, Platzer M, Close TJ, Scholz U, Hansson M, Zhang G, Braumann I, Spannagl M, Li C, Waugh R, Stein N2019A chromosome conformation capture ordered sequence of the barley genomehttps://plants.ensembl.org/Hordeum_vulgare/Info/Annotation/Ensembl Plants, GCA_901482405.1 Chen J, Huang Q, Gao D, Wang J, Lang Y, Liu T, Li B, Bai Z, Luis G, Liang C, Chen C, Zhang W, Sun S, Liao Y, Zhang X, Yang L, Song C, Wang M, ShiJ, Liu G, Liu J, Zhou H, Zhou W, Yu Q, An N, Chen Y, Cai Q, Wang B, Liu B, MinJ, Huang Y, Wu H, Li Z, Zhang Y, Yin Y, Song W, Jiang J, Jackson SA, Wing RA2012Whole-genome sequencing of Oryza brachyantha reveals mechanisms underlying Oryza genome evolutionhttps://plants.ensembl.org/Oryza_brachyantha/Info/Annotation/Ensembl Plants, GCA_0002310 95.2 Kawahara Y, de la Bastide M, Hamilton JP, Kanamori H, McCombie WR, Ouyang S, Schwartz DC, Tanaka T, Wu J, Zhou S, Childs KL, Davidson RM, Lin H, Quesada-Ocampo L, Vaillancourt B, Sakai H, Lee SS, Kim J, Numa H, Itoh T, Buell CR, Matsumoto T2015Improvement of the Oryza sativa Nipponbare reference genome using next generation sequence and optical map datahttps://plants.ensembl.org/Oryza_sativa/Info/AnnotationEnsembl Plants, GCA_001433935.1 Bennetzen JL, Schmutz J, Wang H, Percifield R, Hawkins J, Pontaroli AC, Estep M, Feng L, Vaughn JN, Grimwood J, Jenkins J, Barry K, Lindquist E, Hellsten U, Deshpande S, Wang X, Wu X, Mitros T, Triplett J, Yang X, Ye CY, Mauro-Herrera M, Wang L, Li P, Sharma M, Sharma R, Ronald PC, Panaud O, Kellogg EA, Brutnell TP, Doust AN, Tuskan GA, Rokhsar D, Devos KM2015Reference genome sequence of the model plant Setariahttps://phytozome.jgi.doe.gov/pz/portal.html#!info?alias=Org_SitalicaPhytozome v12.1, AGNK01000000.1 Saski C, Lee SB, Fjellheim S, Guda C, Jansen RK, Luo H, Tomkins J, Rognli OA, Daniell H, Clarke J2017Complete chloroplast genome sequences of Hordeum vulgare, Sorghum bicolor and Agrostis stolonifera, and comparative analyses with other grass genomeshttps://plants.ensembl.org/Sorghum_bicolor/Info/Annotation/Ensembl Plants, GCA_000003195.3 International Wheat Genome Sequencing Consortium (IWGSC), International Wheat Genome Sequencing Consortium (IWGSC)2018Bread wheat variety Chinese Spring for generating IWGSC RefSeq v1.0 assemblyhttps://plants.ensembl.org/Triticum_aestivum/Info/Annotation/Ensembl Plants, GCA_900519105.1 Jiao Y, Peluso P, Shi J, Liang T, Stitzer MC, Wang B, Campbell MS, Stein JC, Wei X, Chin CS, Guill K, Regulski M, Kumari S, Olson A, Gent J, Schneider KL, Wolfgruber TK, May MR, Springer NM, Antoniou E, McCombie WR, Presting GG, McMullen M, Ross-Ibarra J, Dawe RK, Hastie A, Rank DR, Ware D2017Improved maizereference genome with single-molecule technologieshttps://plants.ensembl.org/Zea_mays/Info/AnnotationEnsembl Plants, GCA_000005005.

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
