## [Decision Letter]

Thank you for submitting your article "Two NLR receptors gained high-affinity binding to a fungal effector via convergent evolution of their integrated domain" for consideration by *eLife*. Your article has been reviewed by 3 peer reviewers, one of whom is a member of our Board of Reviewing Editors, and the evaluation has been overseen by Jürgen Kleine-Vehn as the Senior Editor. The following individual involved in review of your submission has agreed to reveal their identity: Pierre-Marc Delaux (Reviewer #2).

Essential revisions:

1. Phylogenetics: From the phylogenetic tree presented in Figure 1A it seems that Pik-1 experienced a duplication before the radiation of the BOP and PACMAD clades, with varying patterns of gene retention/loss (for instance loss of both copies in Brachypodium, loss in one clade for maize) and expansion (massive in wheat for instance in the clade where the fusion with the HMA domain did not occur, not in the other). Please discuss this point the revision, as it may have an important impact: This would support the hypothesis that the HMA integration occurred before the radiation of the PACMAD clade. A better resolved phylogeny is needed to further test this possibility. In that context, the nomenclature should restrict the Pik-1 name to the actual orthologs, changing the number of Pik-1 per species (in panel 1D for instance).

2. Immune recognition of Avr-Pikm-1: If possible, the correlation between evolution of high-affinity binding to AVR-PikD and the ability to induce immune response should be tested in reconstructed ancestral Pikm-1 variants. The presented data demonstrate nicely the gain of high-affinity binding in Pikm-1, but the impact this may have on the actual immunity function was not tested. It would be important to know whether additional mutations were required or not to turn the ancestral Pik1 into a functional Pikm-1 given that it is the basis for the model proposed in Figure 9. Alternatively, as the result of this experiment would not contradict the model even in absence of immune abilities (it would just add one extra step from high-affinity binding to immune function) the authors could propose this second evolutionary scenario as a supplementary figure.

3. Evolutionary model: The authors observe that the inferred ancient integrated domain alleles have low affinity for AVR-PikD. From this they conclude that the domain has recently evolved to recognize an effector different from its ancestral target. But what about recognition of ancient AVR-PikD? How do the authors eliminate this possibility? In an arms-race model, the null expectation would be that the NLR and effector would both change over time continuously (and perhaps indefinitely). The paper would benefit substantially from a more in-depth exploration/discussion of how the current results inform on what is already known of coevolution between NLRs and effectors. We suggest adding a section in the discussion to discuss the arms-race in this macro-evolutionary context.

4. Labeling: The authors work with a number of mutants and do their best to label them clearly throughout, but at some points it is difficult to follow. The figures are already beautifully designed for comprehension but if there is some way to help the reader find key residues/information, particularly in later figures, it would assist reading. There are a few comments to this effect in the detailed Reviewer's comments that the authors can draw from.

Additional Suggestions:

Several other points were raised during the review process that we invite you to review and respond to when you resubmit your manuscript.

*Reviewer #2 (Recommendations for the authors):*

Figure S1: please change the color of the legend for Brachypodium distachyon which look very similar to O. brachyanta

Figure S2: Please provide a.fasta table with the sequenced gene fragments (I do not doubt the results, but this is important for the reproducibility of the work).

Table S4: Please include the bibliographic references describing the published assemblies.

Figure S6: in the legend "Figure S7" should be changed to "Table S7". To improve readability, the branches of the tree and the squares should be aligned.

The nomenclature used for the Pik variants is not consistent throughout the manuscript, please homogenize.

Figure S13: there is an extra "o" in the legend.

L511: "followed by and" → "and".

Figure S16: Only Ponceau and PierceTM used for staining, not CBB as mentioned.

*Reviewer #3 (Recommendations for the authors):*

This is an impressive manuscript that integrates across sub-disciplines in biology and shows compelling results regarding the adaptive evolution of an integrated domain. I detail my two major critiques below, both having to do with the framing of the results.

Overall, I found the interpretation and discussion around the evolutionary analysis to be lacking. The authors observe that the inferred ancient integrated domain alleles have low affinity for AVR-PikD. From this they conclude that the domain has recently evolved to recognize an effector different from its ancestral target. But what about recognition of ancient AVR-PikD? How do the authors eliminate this possibility? In an arms-race model, the null expectation would be that the NLR and effector would both change over time continuously (and perhaps indefinitely). The paper would benefit substantially from a more in-depth exploration/discussion of how the current results inform on what is already known of coevolution between NLRs and effectors.

Lastly, the manuscript is very dense with many experiments and analyses. Only a fraction of these analyses are important to the major inclusions of the manuscript in its current state, and I would encourage the authors to reduce the content of the manuscript substantially for future submissions. In this same vein, I found myself wondering whether this should be two manuscripts-one studying the molecular evolution of the NLR locus and the other a structure-function analysis of the important residues for recognition. Because the paper contains so many different types of data, and so much data, the authors often don't explore their results in detail nor their interpretation. Overall, I liked the paper but found that the paper did not offer a great deal of novel insight into the evolution of NLR-effector interactions. Perhaps with greater focus, the novelty of the findings will become clearer.

---

## [Author Response]

Essential revisions:1. Phylogenetics: From the phylogenetic tree presented in Figure 1A it seems that Pik-1 experienced a duplication before the radiation of the BOP and PACMAD clades, with varying patterns of gene retention/loss (for instance loss of both copies in Brachypodium, loss in one clade for maize) and expansion (massive in wheat for instance in the clade where the fusion with the HMA domain did not occur, not in the other). Please discuss this point the revision, as it may have an important impact: This would support the hypothesis that the HMA integration occurred before the radiation of the PACMAD clade. A better resolved phylogeny is needed to further test this possibility. In that context, the nomenclature should restrict the Pik-1 name to the actual orthologs, changing the number of Pik-1 per species (in panel 1D for instance).

The hypothesis that Pik-1 and Pik-2 duplicated before the radiation of the BOP and PACMAD clades is attractive as it provides a framework for understanding the circumstances of Pik-1 and Pik-2 convoluted evolutionary history and the HMA integration (with one of the duplicated copies fusing to the HMA domain). Although this hypothesis is possible, we feel that given the current resolution of the phylogenetic tree presented in Figure 1A there’s not enough supportive evidence to divide the Pik orthologues into two subgroups. With the current data, the two clades that might have emerged from an early duplication are not well-defined and supported, which is particularly evident in the unrooted tree that can be accessed through an iTOL link provided in the figure legend. In addition, members of both clades are genetically linked with Pik-2, adding another level of complexity. This said, to clarify the reviewer’s point, we incorporated the following sentence to the text: “It is possible that ancestral Pik-1 and Pik-2 experienced a duplication before the radiation of the Poodieae and Panicoideae followed by different patterns of gene loss/retention among grass species, however, a better-resolved phylogeny is needed to test this possibility.” We are excited at the possibility of getting access to high-quality grass genomes in the future and testing this hypothesis by sampling a wider range of grass species and filling the gaps in our phylogeny.

2. Immune recognition of Avr-Pikm-1: If possible, the correlation between evolution of high-affinity binding to AVR-PikD and the ability to induce immune response should be tested in reconstructed ancestral Pikm-1 variants. The presented data demonstrate nicely the gain of high-affinity binding in Pikm-1, but the impact this may have on the actual immunity function was not tested. It would be important to know whether additional mutations were required or not to turn the ancestral Pik1 into a functional Pikm-1 given that it is the basis for the model proposed in Figure 9. Alternatively, as the result of this experiment would not contradict the model even in absence of immune abilities (it would just add one extra step from high-affinity binding to immune function) the authors could propose this second evolutionary scenario as a supplementary figure.

While addressing this comment, we run into several technical problems, including (i) autoactivity of Pikm-1:ancHMA, (ii) perturbed response to AVR-PikD, (iii) reduced protein accumulation levels, and (iv) weak/inconsistent HR. All of these made these additional experiments largely inconclusive. Since these results are inconclusive, we decided not to include them in the final version. We also fear that adding them would affect readability and accessibility of an already dense manuscript. Instead, for transparency, we published all the additional experiments as a Technical Note in Zenodo (https://doi.org/10.5281/zenodo.4770746) to make them available to the community. We included this citation in the Results section entitled “The ANK-VKE mutations confer high-affinity AVR-PikD binding in Pikm-HMA”:

“Next, we aimed to determine whether the gain of AVR-PikD binding translates to an immunoactive Pikm-1:ancHMA by means of HR cell death assay. […] These precluded reliable studying of how the strength of AVR-PikD binding correlates with HR cell death.”

3. Evolutionary model: The authors observe that the inferred ancient integrated domain alleles have low affinity for AVR-PikD. From this they conclude that the domain has recently evolved to recognize an effector different from its ancestral target. But what about recognition of ancient AVR-PikD? How do the authors eliminate this possibility? In an arms-race model, the null expectation would be that the NLR and effector would both change over time continuously (and perhaps indefinitely). The paper would benefit substantially from a more in-depth exploration/discussion of how the current results inform on what is already known of coevolution between NLRs and effectors. We suggest adding a section in the discussion to discuss the arms-race in this macro-evolutionary context.

We selected AVR-PikD precisely because it reflects the ancestral state of all known AVR-Pik alleles (introduction; Bentham et al., 2021; Kanzaki et al., 2012; Longya et al., 2019). This should exclude the possibility that AVR-PikD recognition is a result of an ongoing arms race between the AVR-Pik effectors and the Pik receptors. We have now also included this information in the Results section (in addition to the introduction) entitled: “Reconstructed ancHMAs exhibit weaker association with AVR-PikD compared to modern Pikp-HMA”.

This said, it is of course possible that the integrated HMA domain might have recognised an AVR-PikD progenitor that is now extinct. We now briefly discuss this possibility in paragraph #5 of the discussion, where we also provide a more in-depth discussion about different models of effector–NLR evolution. We have also cited Bentham et al., 2021—a newly released preprint that adds to our understating of the AVR-Pik evolution and AVR-Pik–related effectors.

4. Labeling: The authors work with a number of mutants and do their best to label them clearly throughout, but at some points it is difficult to follow. The figures are already beautifully designed for comprehension but if there is some way to help the reader find key residues/information, particularly in later figures, it would assist reading. There are a few comments to this effect in the detailed Reviewer's comments that the authors can draw from.

Following reviewer’s comment, we included the amino acid alignments that highlight key residues to figures 5, 8, and 9. We also added additional descriptions to figures 4 and 7.

Reviewer #3 (Recommendations for the authors):This is an impressive manuscript that integrates across sub-disciplines in biology and shows compelling results regarding the adaptive evolution of an integrated domain. I detail my two major critiques below, both having to do with the framing of the results.Overall, I found the interpretation and discussion around the evolutionary analysis to be lacking. The authors observe that the inferred ancient integrated domain alleles have low affinity for AVR-PikD. From this they conclude that the domain has recently evolved to recognize an effector different from its ancestral target. But what about recognition of ancient AVR-PikD? How do the authors eliminate this possibility? In an arms-race model, the null expectation would be that the NLR and effector would both change over time continuously (and perhaps indefinitely). The paper would benefit substantially from a more in-depth exploration/discussion of how the current results inform on what is already known of coevolution between NLRs and effectors.

Response included in the essential revisions.

Lastly, the manuscript is very dense with many experiments and analyses. Only a fraction of these analyses are important to the major inclusions of the manuscript in its current state, and I would encourage the authors to reduce the content of the manuscript substantially for future submissions. In this same vein, I found myself wondering whether this should be two manuscripts-one studying the molecular evolution of the NLR locus and the other a structure-function analysis of the important residues for recognition. Because the paper contains so many different types of data, and so much data, the authors often don't explore their results in detail nor their interpretation. Overall, I liked the paper but found that the paper did not offer a great deal of novel insight into the evolution of NLR-effector interactions. Perhaps with greater focus, the novelty of the findings will become clearer.

Although we appreciate that the manuscript is quite long and dense, we feel that splitting it into two separate manuscripts would take away from the overall evolutionary perspective. In its current form, the paper explores the dynamics of Pik-1 evolution at different timescales.